# Spinal Muscular Atrophy: The Past, Present, and Future of Diagnosis and Treatment

**DOI:** 10.3390/ijms241511939

**Published:** 2023-07-26

**Authors:** Hisahide Nishio, Emma Tabe Eko Niba, Toshio Saito, Kentaro Okamoto, Yasuhiro Takeshima, Hiroyuki Awano

**Affiliations:** 1Faculty of Rehabilitation, Kobe Gakuin University, 518 Arise, Ikawadani-cho, Nishi-ku, Kobe 651-2180, Japan; 2Laboratory of Molecular and Biochemical Research, Biomedical Research Core Facilities, Juntendo University Graduate School of Medicine, 2-1-1 Hongo, Bunkyo-ku, Tokyo 113-8421, Japan; niba@juntendo.ac.jp; 3Department of Neurology, National Hospital Organization Osaka Toneyama Medical Center, 5-1-1 Toneyama, Toyonaka 560-8552, Japan; saito.toshio.cq@mail.hosp.go.jp; 4Department of Pediatrics, Ehime Prefectural Imabari Hospital, 4-5-5 Ishi-cho, Imabari 794-0006, Japan; kentaro206@gmail.com; 5Department of Pediatrics, Hyogo Medical University, 1-1 Mukogawacho, Nishinomiya 663-8501, Japan; ytake@hyo-med.ac.jp; 6Organization for Research Initiative and Promotion, Research Initiative Center, Tottori University, 86 Nishi-cho, Yonago 683-8503, Japan; awano@tottori-u.ac.jp

**Keywords:** spinal muscular atrophy, classification, *SMN1*, *SMN2*, antisense oligonucleotides, gene therapy, low-molecular-weight compounds, newborn screening

## Abstract

Spinal muscular atrophy (SMA) is a lower motor neuron disease with autosomal recessive inheritance. The first cases of SMA were reported by Werdnig in 1891. Although the phenotypic variation of SMA led to controversy regarding the clinical entity of the disease, the genetic homogeneity of SMA was proved in 1990. Five years later, in 1995, the gene responsible for SMA, *SMN1*, was identified. Genetic testing of *SMN1* has enabled precise epidemiological studies, revealing that SMA occurs in 1 of 10,000 to 20,000 live births and that more than 95% of affected patients are homozygous for *SMN1* deletion. In 2016, nusinersen was the first drug approved for treatment of SMA in the United States. Two other drugs were subsequently approved: onasemnogene abeparvovec and risdiplam. Clinical trials with these drugs targeting patients with pre-symptomatic SMA (those who were diagnosed by genetic testing but showed no symptoms) revealed that such patients could achieve the milestones of independent sitting and/or walking. Following the great success of these trials, population-based newborn screening programs for SMA (more precisely, *SMN1*-deleted SMA) have been increasingly implemented worldwide. Early detection by newborn screening and early treatment with new drugs are expected to soon become the standards in the field of SMA.

## 1. Introduction

Spinal muscular atrophy (SMA) is a lower motor neuron disease with autosomal recessive inheritance that results in progressive proximal muscle weakness and skeletal muscle atrophy. The incidence of SMA is approximately 1 in 10,000 to 20,000 live births, and the carrier frequency is 1/40 to 1/70 in the general population [1]. SMA is believed to be the leading genetic cause of infant mortality [2].

In 1891, the Austrian neurologist Guido Werdnig first reported two cases of SMA with common features [3]. It is worthy of notice that their phenotypes were consistent with those of SMA type II, according to the current classification of SMA (we will discuss it later). However, many cases with phenotypes different from those described by Werdnig have since been reported.

Until the 1980s, SMA was divided into several clinical groups. Among these, the two major groups were acute infantile SMA (also termed SMA type I or Werdnig–Hoffmann disease) and chronic childhood SMA (also termed SMA types II and III, arrested Werdnig–Hoffmann disease, Kugelberg–Welander disease, or chronic generalized SMA) [4].

In 1990, the disease loci in these patients with SMA were mapped to the same region in chromosome 5q13, and they were grouped into one clinical entity (5q-SMA, hereafter simply referred to as “SMA”) [5,6,7,8]. After SMA was recognized as a monogenic disease with a wide range of clinical manifestations, the most severe fetal-onset form and the mildest adult-onset form (type 0 and type IV, respectively) were also considered forms of SMA [9,10].

SMA is currently divided into five subtypes [11]: type 0 (the most severe form with onset in the prenatal period and development of severe respiratory problems after birth), type I (Werdnig–Hoffmann disease; a severe form with onset before 6 months of age and the inability to sit unsupported), type II (Dubowitz disease; an intermediate form with onset before 18 months of age and the ability to sit unaided, but not to stand or walk), type III (Kugelberg–Welander disease; a mild form with onset after 18 months of age and the ability to stand and walk unaided), and type IV (the mildest form with onset after 30 years of age).

The gene responsible for SMA, *SMN1*, was identified in 1995 [12]. The SMA-modifier gene, *SMN2*, was also identified at the same time as *SMN1* [12]. The discovery of these two genes led to the next stage of SMA research. Before the identification of responsible gene for SMA, *SMN1*, histological findings of the biopsied muscles were very important for the diagnosis of SMA [13]. SMA was definitely diagnosed based on the characteristic findings of denervation in the muscle specimen; grouped atrophy of muscle fibers and/or angulated muscle fibers. However, after the discovery of the *SMN1* gene, diagnostic procedures have been completely changed [11]. Muscle biopsy has no longer been performed to diagnose SMA. Instead, SMA has been diagnosed based on the genetic testing of *SMN1* deletion or mutation. 

Then we entered the era of drug development for the patients with SMA. Dubowitz stated in his essay, “We live in exciting times, and it has been a tremendous privilege to witness the remarkable growth in interest in the neuromuscular disorders over the past half-century and the quantum leap since mid-1990s in resolving the complex genetic background of SMA, now on the cusp of potential therapy” [14]. In 2016, nusinersen was the first drug approved for treatment of SMA by the Food and Drug Administration (FDA) in the United States (US) [15]. Two other drugs were subsequently approved by the FDA: onasemnogene abeparvovec and risdiplam [15].

SMA has been recognized as an incurable disease since its first description at the end of the 19th century; it remains incurable even with the introduction of new drugs at the beginning of the 21st century. Although these new drugs have markedly improved the symptoms and prognosis of SMA, they have not completely cured the disease. Nevertheless, as we look back over the accomplishments of 132 years of SMA research, we take pride in many of those great moments in history. In this review, we trace the history of SMA research and describe the advancements in clinical classification, molecular genetic analysis, and treatment of this disease.

## 2. Single-Gene Disorder with Phenotypic Variation

### 2.1. First Case Report of SMA

The history of SMA research may be divided into three periods (Figure 1). The first period (1891–1994) began with the first report of patients with SMA, and it included classification of clinical subtypes and chromosomal mapping of the SMA locus. The second period (1995–2015) began with cloning of the *SMN* genes, and it included drug repositioning and the development of new drugs with the aim of curing SMA. The third period (2016–present) began with introduction of FDA-approved drugs for SMA into the clinical setting, and it includes newborn screening for early diagnosis of SMA. In this article, Section 2 describes the first period, Section 3, Section 4 and Section 5 describe the second period, and Section 6, Section 7 and Section 8 describe the third period.

In 1891, Werdnig published a historic article titled “Zwei frühinfantile hereditäre Fälle von progressiver Muskelatrophie unter dem Bilde der Dystrophie, aber auf neurotischer Grundlage (Two early childhood hereditary cases of progressive muscular atrophy with the appearance of dystrophy but on a neurogenic basis)” [3]. The history of SMA began with this case report of two brothers who had progressive muscular atrophy. Both brothers were healthy until the age of 10 months. Based on the description that they sat holding the feeding bottle with both hands, the brothers may have achieved the milestone of independent sitting. The younger brother achieved the milestone of standing, although whether this was independent standing remains unclear. After the age of 10 months, however, the brothers rapidly lost the motor function of their extremities. The author diagnosed their muscle impairment as Leyden–Moebius dystrophy, which is now considered linked to limb-girdle muscular dystrophy and is characterized by muscle weakness that is dominant in the proximal portions of the limbs. Autopsy of the brothers revealed degeneration of the anterior horn cells of the spinal cord and simple atrophy of the muscles.

Soon after publication of the above report, Hoffman (1893) and Thomson and Bruce (1893) described patients with a clinical course similar to that of Werdnig’s patients [16,17]. All of these patients are now considered to have had intermediate SMA or SMA type II. Although Werdnig’s and Hoffmann’s names have been associated with the severe form of SMA, as a matter of fact, they described the intermediate form of SMA (or SMA type II).

### 2.2. Phenotypic Variation of SMA

#### 2.2.1. Severe Phenotype (SMA Type I, Werdnig–Hoffmann Disease)

In 1902, Beevor described a male patient with a more severe form of SMA [18]. The patient was the eighth child in a family with eight children, and he was the fourth affected child. The first affected sibling (sister) was noticed to be paralyzed all over at the end of the first month of life and died before 4.5 months of age. The second affected sibling (sister) developed symptoms at 6 months of age, and died at 8 months of age. The third affected sibling (gender unknown) developed symptoms at 6 weeks of age and died at 7 months of age. The patient was noticed to be paralyzed all over, excepting the diaphragm, before 5 weeks of age, and died after living eight weeks. The patient was considered to have congenital SMA (family type). The post-mortem examination of the patient showed remarkable atrophic changes in the cells of anterior horns of the spinal cord, and typical findings of grouped atrophy of the muscle fibers. If these affected siblings were born into the world today, they would be diagnosed with SMA type I. The author recognized the differences between his case and Werdnig’s and Hoffmann’s cases.

According to historical studies by Dubowitz [14,19], Sylvestre had already described an early infantile form of SMA before Beevor reported it. In 1899, Sylvestre presented a case of a 2-month-old male infant who had exhibited flaccid paralysis of all four limbs and trunk since birth and showed weakness of the intercostals with sparing of the diaphragm [20]. The patient was the sixth child in a family with six children, and he was the third affected child. The first affected child (brother) died at 3 months of age, and the second affected child (sister) died at 5 months of age. Although Sylvestre’s report contained no pathological morphological findings, the symptoms of the patient appear to be compatible with the severe form of SMA, SMA type I. The author also recognized the differences between his case and Hoffmann’s cases.

These facts raise the question of how this severe, early infantile form of SMA has become known as Werdnig–Hoffmann disease. A reasonable explanation is as follows. In 1900, Oppenheim reported infants who were born with severe muscular hypotonia [21]. From then until the mid-20th century, the term “amyotonia congenita (Oppenheim)” was used by many pediatricians to describe all cases of early infantile hypotonia and muscle weakness [22]. Brandt reported in 1950 that in 38 out of 112 cases of infantile SMA, the original diagnosis had been amyotonia congenita [23].

In the midst of the diagnostic confusion between “amyotonia congenita“ and “infantile SMA”, Greenfield and Stern reported that the pathological findings of some infants diagnosed with “amyotonia congenita” were identical to the pathological findings of cases described by Werdnig and Hoffmann [24]. Their report may have led physicians and researchers to refer to SMA that develops in early infancy as “amyotonia congenita” or “Werdnig–Hoffmann disease”. The neonatal hypotonia reported as “amyotonia congenita” was later clarified to be a symptom present in various conditions [25], and the term “amyotonia congenita” as a clinical entity was therefore discarded [26].

However, the link between early infantile SMA and Werdnig–Hoffmann disease remains today. This may be because Werdnig and Hoffmann attracted attention as the discoverers of the disease causing hypotonia in early infancy (then many cases might have been diagnosed as amyotonia congenita), which was a hot topic in the early 20th century.

#### 2.2.2. Intermediate Phenotype (SMA Type II, Dubowitz Disease)

Intermediate SMA was first described by Werdnig, Hoffmann, and Thomson and Bruce, as described above, but it is currently known as Dubowitz disease [27].

Dubowitz detailed the phenotype of intermediate SMA in 1964 [28]. He reported a series of 12 patients with the intermediate form in his 1964 article and abstracted the following characteristic features of intermediate SMA from them: generalized weakness in the trunk and limb muscles, ability to sit unaided, inability to stand and walk, progressive scoliosis, contracture in the hips and knees, and joint laxity of the hands and fingers. Dubowitz also described the apparent variation in severity among affected siblings within the same family, suggesting the presence of different grades of severity (or phenotypic variation in severity) of one basic disease process, rather than separate disease entities.

#### 2.2.3. Mild Phenotype (SMA Type III, Kugelberg–Welander Disease)

In 1956, Kugelberg and Welander documented 12 juvenile patients with a milder form of SMA [29]. Patients in their 40s and 50s were included in their report, suggesting that these patients lived longer than patients with severe SMA in the previous studies. In addition, the patients’ medical history showed that their age at symptom onset ranged from 2 to 17 years, suggesting a long period during which independent walking ability was maintained as well as a slower and more favorable course than that of severe and intermediate SMA. 

In each of these patients, the disorder had originally been diagnosed as muscular dystrophy. However, electromyography and muscle biopsy presented evidence of lower motor neuron damage, suggesting that the muscular disorder was secondary to a lesion of spinal motor neurons. The authors of this report were eponymized in Kugelberg–Welander disease to refer to mild SMA (SMA type III).

### 2.3. Single-Gene Disorder

#### 2.3.1. SMA Locus in Chromosome 5q13

The discovery of phenotypic variation led to controversy regarding whether this variation reflects genetic heterogeneity [14]. Assuming SMA is a group of genetically heterogeneous diseases, the existence of at least two or three genetically distinct diseases should be expected. However, after genetic homogeneity was proved in 1990, the continuous spectrum of SMA became accepted by physicians worldwide.

In 1990, Gilliam et al. in the US and Melki et al. in France mapped the gene locus of chronic SMA (intermediate and mild SMA) and acute SMA (severe SMA) to the same region of chromosome 5q. According to the report by Gilliam et al., the locus of SMA was chromosome 5q11.2-13.3 [5,6]. According to the report by Melki et al., it was chromosome 5q12-14 [7,8].

#### 2.3.2. Current Classification of SMA

According to the classification issued at the 134th European Neuromuscular Centre (ENMC) International Workshop 2005 (Naarden, The Netherlands), SMA type I is subdivided into types Ia, Ib, and Ic [30]. Type Ia is the most severe neonatal form of type I. Joint contracture and extreme hypokinesia are observed at birth, and artificial ventilation is required during the neonatal period. Type Ib is a common form of type I in which head control is difficult. Type Ic is the mildest form of type I; affected patients achieve head control and can assume a sitting position with support. Type II is not subclassified. Type III is subdivided into type IIIa (age at onset: <3 years) and type IIIb (age at onset: >3 years) This subclassification is based on the previous study focusing on the difference between them in the age at which walking ability is lost [31].

At the 209th ENMC International Workshop 2014 (Heemskerk, The Netherlands), the items “type 0” and “type IV” were finally adopted into the disease classification [32]. Type 0 is a prenatal-onset form of SMA. In 1999, MacLeod et al. reported cases of prenatal-onset SMA [33], and Dubowitz proposed classifying the most severe form of prenatal-onset SMA as “type 0” [9]. Type IV is an adult-onset form of SMA. In 1995, Clermont et al. first reported an adult-onset SMA patient with deletion of the SMA-causative gene, *SMN1* [10]. In the classification based on the 209th ENMC International Workshop 2014, the age at onset of type IV was >35 years. Oskoui et al. proposed that the age at onset of type IV was >21 years [34].

However, a classification containing a detailed subclassification may be difficult to remember. In addition, clinicians cannot always differentiate between neighboring subtypes (e.g., type 0 vs. Ia and type IIIb vs. IV). Arnold et al. proposed a simple classification without a detailed subclassification (Table 1) [11].

## 3. The SMA-Causative Gene: *SMN1*

### 3.1. The SMN1 and SMN2 Genes

In 1995 (five years after the SMA locus was mapped to chromosome 5q13), two candidate genes responsible for SMA were cloned from the same region: the Survival of Motor Neuron 1 gene (*SMN1* or *^T^BCD541*) and the telomeric Neuronal Apoptosis Inhibitory Protein (*telomeric NAIP*) gene [12,35].

The genomic organization of the SMA locus in the chromosome 5q13 region is complicated. A 500-kb duplication is present in the 5q13 region, and two homologous genes (paralogs) are aligned on the telomeric side and the centromeric side [12]. There are two models of alignment of the homologous genes, the inverted and tandem duplication models [12,36]. In both models, *SMN1* and *telomeric NAIP* are present on the telomeric side, while the centromeric homologous genes, the Survival of Motor Neuron 2 gene (*SMN2* or *^C^BCD541*: a homologous gene to *SMN1*) and *centromeric NAIP* (a homologous gene to *telomeric NAIP*) are present on the centromeric side [12]. The *SMN1* and *SMN2* genes are shown according to two models in Figure 2.

According to the first report of *SMN1* by Lefebvre et al. [12], complete absence of *SMN1* (or homozygous *SMN1* deletion) was found in 226 (98.7%) of 229 patients with SMA irrespective of their clinical subtypes, and an intragenic *SMN1* mutation (a small mutation in *SMN1*) was found in 3 (1.3%) of these 229 patients. Based on these findings, it has been considered that *SMN1* mutation causes SMA or that *SMN1* is an SMA-causative gene.

The *SMN1* gene is approximately 20 kb in length and consists of nine exons (exons 1, 2a, 2b, 3, 4, 5, 6, 7, and 8) [37]. The translation initiation codon (ATG) and termination codon (TAA) are located within exons 1 and 7, respectively [12,37]. The gene encodes the full-length SMN protein (hereafter, full-length SMN protein is referred to as only “SMN protein” unless otherwise stated), which comprises 294 amino acids and is 38 kDa in length [12,38].

### 3.2. The SMN Protein

#### 3.2.1. Expression in Tissues and Subcellular Localization 

Lefebvre et al. reported that the *SMN1* gene product, SMN protein, was markedly decreased in lymphoblastoid cell lines and tissues (spinal cord and liver) from patients with SMA [39]. It is now well-known that SMN protein is ubiquitously expressed in all cells and tissue types, not only motor neurons [11]. 

Liu and Dreyfuss originally described the subcellular localization of the SMN protein in HeLa cells [40]. SMN protein is found in discrete nuclear structures and in the cytoplasm. Liu and Dreyfuss termed SMN protein-containing nuclear structures “gems” in reference to Gemini of Cajal bodies (or coiled bodies) because they are similar in number (2–6) and size (0.1–1.0 μm) to Cajal bodies and are frequently found near or associated with Cajal bodies. The model of the SMN protein complex shuttling between the nucleus and cytoplasm is related to small nuclear ribonucleoprotein (snRNP) biogenesis [41,42].

#### 3.2.2. High SMN Protein Expression in the Fetal Period

Burlet et al. compared SMN protein levels in fetal and postnatal controls using skeletal muscle, heart, kidney, thymus, pancreas, and lung [43]. According to their study, the SMN protein level in the brain was higher in fetal controls than in postnatal controls.

Ramos et al. also quantified SMN protein levels in spinal cord samples isolated during expedited autopsies [44]. They demonstrated that SMN protein expression was restricted at relatively low levels in controls after 3 months of life. Their study results showed that SMN protein levels varied widely; the median SMN protein level was 2.3-fold higher in prenatal controls than in postnatal controls aged < 3 months (early postnatal stage) and 6.5-fold higher than in postnatal controls aged 3 months through 14 years (late postnatal stage). 

These findings suggested that SMN protein level is high in the fetal period but rapidly decreases after birth.

#### 3.2.3. Various Functions of SMN Protein

In 1997, only two years after the discovery of *SMN1*, Dreyfuss et al. demonstrated that the SMN protein is associated with snRNP biogenesis and splicing of pre-mRNA [45,46]. This finding may be consistent with the ubiquitous expression of SMN protein. This finding also raised a question: Why does loss of (more precisely, a decrease in) the ubiquitously expressed SMN protein result in selective degeneration of motor neurons in patients with SMA? Monani discussed two possible contrasting views: (1) SMA is a consequence of a defect in snRNP biogenesis and pre-mRNA splicing or (2) SMA is a consequence of a defect in the motor neuron-specific function of SMN protein [47]. Here, Monani et al. presented the idea of considering the *SMN1* gene as a housekeeping gene and the SMN protein as a housekeeping gene product. Burghes and Beattie also presented two hypotheses: (1) loss of the function of SMN protein in snRNP assembly causes an alteration in the splicing of a specific gene (or genes); and (2) SMN protein is critical for the transport of mRNA in neurons, and disruption of this function results in SMA [48].

Generally, SMN protein domains are specifically bound to their partner proteins. The most studied domains are the Tudor domain (encoded by exon 3) and the C-terminal domain of SMN protein (encoded by exons 6 and 7) (Figure 3). The Tudor domain of SMN protein is responsible for an interaction with coilin protein, a marker of Cajal bodies [49]. The domain also binds to the C-terminal arginine- and glycine-rich tails of Sm core proteins, which contain symmetrical dimethylated arginine residues, thereby facilitating the assembly of spliceosomes [50,51]. The YG box is a tyrosine/glycine-rich region in the C-terminus of SMN protein that facilitates oligomerization of SMN protein by formation of the glycine zipper structure [52]. However, while these findings related to the Tudor and C-terminal domains are still important in considering the functions of SMN protein, they may not answer the questions above.

Many studies of the partner proteins bound to the domains have been performed to elucidate the functions of SMN associated with pathogenesis of SMA. Based on those studies, SMN protein is now known to be involved in a wide variety of cellular functions: small nuclear ribonucleoprotein (snRNP) biogenesis and splicing of pre-mRNA, mRNA trafficking and local translation, cytoskeletal dynamics, endocytosis and autophagy, mitochondrial dysfunction and bioenergetic pathways, ubiquitin–proteasome system [53,54]. We should not think of SMN as simply a housekeeping gene product, nor should we consider that the housekeeping gene product has only one function. It remains difficult to elucidate the full picture of all SMN functions and identify those functions related to the pathogenesis of SMA.

#### 3.2.4. SMN Protein in Tissues Associated with SMA Pathology

In order to elucidate the pathogenesis of SMA, research is progressing to seek for SMN-binding proteins involved in the basic functions of motor neurons, neuromuscular junctions, and skeletal muscle. Here, we would like to introduce some recent research results.

Regarding the motor neurons, Rossoll et al. indicated that a complex of Smn protein (or murine protein corresponding to human SMN protein) with its binding partner hnRNP R interacts with β-actin mRNA and translocates to axons and growth cones of motor neurons [55]. There was a series of reports showing that Smn proteins plays a role in axonal RNA metabolism. Fallini et al. pointed out that SMN protein in axonal transport granules and its interaction with numerous mRNA-binding proteins are not related to snRNP biogenesis or splicing regulation [56]. 

Regarding the neuromuscular junction (NMJ), a study using SMA model mice demonstrated impairment of maturation in acetylcholine receptor (AChR) cluster formation, which was reflected in functional deficits at the NMJ [57]. This finding was thought to be associated with axonal trafficking defects in motor neurons [56]. On the other hand, endocytosis impairment in synaptic transmission at NMJs has recently been attracting attention [58,59]. Hosseinibarkooie et al. found that the plastin 3 protein (PLS3) overexpression rescues survival and motoric abilities in severe SMA model mice [58]. PLS3 is an F-actin-binding protein that is involved in many cellular processes including endocytosis. Based on these findings, they suggested that disturbed endocytosis might be a key cellular mechanism underlying impaired neurotransmission and NMJ maintenance in SMA. Riessland et al. also reported that neurocalcin delta (NCALD) suppression may be a protective modifier of SMA [59]. NCALD is a calcium-dependent negative regulator protein of endocytosis. They discovered the relationship between NCALD and SMA phenotype using genetic linkage analysis in SMA families. PLS3 and NCALD as SMA phenotype modifiers will be revisited later. Recently, Kim et al. used SMA model mice to demonstrate that SMN is associated with the SNARE complex assembly [60]. Disrupted SNARE complexes may dysregulate neurotransmission at NMJs. 

Regarding the muscle, the pathologic change in SMA has also come to be studied centering on the SMN protein [61]. Kim et al. studied the selective depletion of SMN protein in skeletal muscle [62]. Although a disease phenotype was not immediately obvious, persistent low levels of the protein eventually resulted in muscle fiber defects, NMJ abnormalities, poor motor performance, and premature death. Recently, Ikenaka et al. found that the down-regulation of SMN causes mitochondrial dysfunction and subsequent cell death in in vitro models of skeletal myogenesis with both a murine C2C12 cell line and human induced pluripotent stem cells [63]. They reported that during myogenesis, a decrease in SMN protein may reduce the production of myoblast determination protein 1 (MYOD1), microRNA (miR)-1 and miR-206, resulting in mitochondrial dysfunction. Their finding suggested that the SMN-miR axis is essential for myogenic metabolic maturation.

### 3.3. Another Candidate SMA-Causative Gene: Telomeric NAIP

In a study by Roy et al., complete absence of *telomeric NAIP* (homozygous *telomeric NAIP* deletion) was found in half of patients with SMA type I. By contrast, the complete absence of *telomeric NAIP* is rarely observed in patients with SMA types II and III [35].

The *telomeric NAIP* gene belongs to the inhibitor of the apoptosis gene family. Its gene product, the NAIP protein, inhibits the action of caspase and suppresses cell apoptosis [64]. Diseases associated with *telomeric NAIP* include cerebral injury; post-ischemic cell death in the brain is suppressed by NAIP protein [65].

How NAIP protein deletion contributes to lower motor neuron dysfunction and loss in humans remains unknown. Although *telomeric NAIP* could be associated with the severity of SMA, it will not be further discussed in this review article.

## 4. The SMA-Modifier Gene: *SMN2*

### 4.1. The SMN2 Gene

*SMN1* (or *^T^BCD541*) is present on the telomeric side, while the centromeric homologous gene is *SMN2* (or *^C^BCD541*) [12] (Figure 2). *SMN1* and *SMN2* have almost the same nucleotide sequence. A limited number of nucleotide differences are present in the entire gene, and the only single-nucleotide difference in the coding region is at the sixth nucleotide position in exon 7 (c.840C in *SMN1* and c.840T in *SMN2*) [12,66]. The C-to-T transition in *SMN2* exon 7 is a synonymous substitution that does not cause an amino acid change; *SMN2* is able to produce the functional SMN.

Lefebvre et al. reported in their *SMN1* discovery paper that absence of *SMN2* was observed in 5% of control individuals and in 0% of patients with SMA [12]. To date, no patients with SMA carrying 0 copies of *SMN1* and 0 copies of *SMN2* have been reported. According to experimental results using SMA model mice, *Smn* (−/−) knockout, is embryonically lethal, but introduction of human *SMN2* rescues the embryo [67,68]. Thus, *SMN2* may produce some amount of functional SMN protein in the embryonic stage of patients with SMA, rescuing the embryonic or fetal death.

### 4.2. Alternative Splicing

Notably, the presence of *SMN2* does not fully compensate for the absence of *SMN1* in patients with SMA. Although the C-to-T transition in *SMN2* is a synonymous mutation, it does not mean that *SMN1* and *SMN2* work in the same way.

*SMN1* produces full-length transcripts, and *SMN2* primarily produces mRNA lacking exon 7 because the C-to-T transition (or C-to-U transition in pre-mRNA) of *SMN2* exon 7 inhibits alternative splicing of exon 7, leading to incomplete inclusion of the exon [69] (Figure 4). The major product of *SMN2* is truncated mRNA lacking exon 7 (Δ7-*SMN2* mRNA), and the minor product is fulllength *SMN2* mRNA. Thus, the major protein product is truncated SMN from Δ7-*SMN2* mRNA, and the minor protein product is full-length SMN protein from full-length *SMN2* mRNA.

As calculated using the blood data of patients with SMA [70,71], *SMN2* produces about 80% to 90% Δ7-*SMN* mRNA and 10% to 20% full-length *SMN* mRNA. *SMN2* is unable to produce sufficient full-length *SMN* mRNA, resulting in a low level of the full-length SMN protein. However, at the protein level, Δ7-SMN protein is an unstable and almost undetectable protein [72,73]. In addition, a feedback loop regulates splicing of the *SNN2* transcript: low levels of full-length SMN protein may exacerbate *SMN2* exon 7 skipping, leading to a further reduction in full-length SMN protein [74]. This is why the presence of *SMN2* does not fully compensate for the absence of *SMN1* in patients with SMA. Even so, an increase in the *SMN2* copy number may produce more full-length SMN protein, which would ameliorate the SMA phenotype. Jodelka et al. suggested that a modest increase in SMN protein abundance may cause a disproportionately large increase in SMN protein expression [74]. The correlation between the *SMN2* copy number and SMA phenotype is discussed later in this review article.

This leads to the question of how the C-to-T transition of *SMN2* exon 7 affects alternative splicing of exon 7. Cartegni and Krainer proposed a theory of exonic splicing enhancer (ESE) motif disruption by the C-to-T transition [75]. The ESE motif is a cis-acting element that facilitates inclusion of the index exon along with a trans-acting factor called SF2/ASF protein. Kashima and Manley proposed a theory of exonic splicing silencer (ESS) motif creation by the C-to-T transition [76]. The ESS motif is a cis-acting element that facilitates exclusion of the index exon along with a trans-acting factor called heterogeneous nuclear RNP (hnRNP) A1/2. Finally, the C-to-T transition changes the condition of U2AF binding and U2 snRNP recruitment on the 3′-splice site [77]. Defective *SMN2* splicing may reflect the fine balance between antagonistic splicing factors and exonic elements in a disease context [78].

Full-length SMN protein is known to oligomerize and form a multimeric protein complex. However, it is difficult for Δ7-SMN protein to oligomerize [79]. SMN oligomerization defects are correlated with SMA severity [79]. Using pulse-chase analysis, Burnett et al. explored the stability and degradation of full-length SMN and Δ7-SMN proteins. SMN protein is degraded by the ubiquitin proteasome system, and the Δ7-SMN protein has a two-fold shorter half-life than full-length SMN protein within cells [80]. In addition, when mutations inhibit SMN protein complex formation with partner proteins, the stability of the SMN protein may decrease [81].

### 4.3. Phenotype Modification by SMN2

#### 4.3.1. *SMN2* Copy Number

The C-to-T transition in *SMN2* is a synonymous substitution, as mentioned above. This means that *SMN2* can produce the same full-length SMN as produced by *SMN1*. Velasco et al. found a higher ratio of *SMN2/SMN1* gene dosage in the parents of patients with SMA type II and III than in the parents of those with type I [82]. Based on this finding, they presented the hypothesis that the existence of multiple copies of the *SMN2* gene may compensate for the deletion of the *SMN1* gene, modifying the severity of SMA. This hypothesis has been supported by many researchers [83,84].

Arnold et al. simplified the reported data on the correlation between phenotype and *SMN2* copy number [11] (Table 1); at least one copy of *SMN2* is required to develop SMA (or prevent embryonic death), and infants with the most severe form of disease (type 0) usually have only one copy. Infants with SMA type I usually have two or three copies of *SMN2*. SMA type II is usually associated with three copies. Patients with SMA type III have three or four copies, and patients with SMA type IV usually have at least four copies. However, in the real world, many patients clinically diagnosed with SMA type IV are not tested for the *SMN1* and *SMN2* genes and may have other genetic abnormalities.

Notably, the correlation between phenotype and *SMN2* copy number is not absolute. Some patients with two *SMN2* copies present with milder SMA phenotypes, whereas some with three copies of the gene have been described as having type I [84,85]. These findings suggested that the numbers of *SMN2* copies are not functionally equivalent among patients with SMA [85], or other genetic modifiers outside of *SMN2* may contribute to the progression of the disease [11].

The inverse correlation between the *SMN2* copy number and SMA severity is applicable to patients with complete deletion of *SMN1*. However, it should also be noted that the *SMN2* copy number does not necessarily correlate with severity in patients with intragenic *SMN1* mutations [86,87].

#### 4.3.2. Gene Conversion Event from *SMN1* to *SMN2*

In our study, the mean *SMN2* copy number in patients with SMA (including all subtypes) was higher than that in control subjects [88]. By contrast, the mean total *SMN* (*SMN1* and *SMN2*) copy number in patients with SMA was lower than that in control subjects [88]. These findings suggest that both *SMN1* deletion and *SMN2* generation occur in patients with SMA. We must therefore consider the mechanism underlying the decreased *SMN1* copy number and increased *SMN2* copy number in patients with SMA, especially patients with three or four copies (often observed in SMA types III and IV).

Lefebvre et al. reported in their *SMN1* discovery paper that amplification of exon 7 to exon 8 in a patient with SMA type III resulted in a product carrying the *SMN2* exon 7 and *SMN1* exon 8 sequences, suggesting the possibility of a gene conversion event in this patient [12]. Since then, many studies on the hybrid *SMN* gene (*SMN2* exon 7 and *SMN1* exon 8) have been reported [89,90]. Such a hybrid gene is apparently far from rare and may partly explain the frequently observed *SMN1* deletions in patients with SMA. The hybrid *SMN* gene may be a partial form of gene conversion.

Based on the gene conversion theory, the mechanism underlying how the *SMN1* copy number is decreased and the *SMN2* copy number is increased in patients with SMA is now explained by gene deletion of *SMN1* and/or gene conversion of *SMN1* to *SMN2* in the telomeric part of the SMA locus of chromosome 5q13 [82,91,92].

The possibility of segmental duplication must also be addressed. The increase in the *SMN2* copy number can be explained by segmental duplication because *SMN2* is a paralogous gene of *SMN1*. However, the copy number of neighboring genes of *SMN2* in the centromeric part of the SMA locus is not increased, denying the possibility of segmental duplication [88].

#### 4.3.3. Variation of *SMN* Gene Copy Numbers in Different Populations

In 2001, Rochette et al. reported that *SMN2* is human-specific and not found in chimpanzees [93]. According to their report, (1) after primates and rodents diverged 750–1100 million years ago, duplication of the *SMN* gene occurred in primates, and (2) humans and chimpanzees diverged 5–7 million years ago. After the latter evolutionary divergence, *SMN2* has emerged in humans. In 2017, Dennis et al. also reported the study results of human-specific segmental duplications from the perspective of molecular evolution. They dated the emergence of the *SMN2* gene to 3 million years ago [94]. In any case, *SMN2* emerged from *SMN1* by segmental duplication millions of years ago. However, once *SMN2* is created, variations in *SMN2* copy number may be due to deletion or gene conversion events involving the genes.

According to the study on prevalence or incidence by SMA subtype in different populations, the frequencies of patients with type I and patients with types II-III vary in different ethnic groups [1]. The frequencies of patients with different subtypes may reflect the higher or lower copy number of *SMN2*. The higher or lower copy number of *SMN2* may be the result of deletion or gene conversion events. In the general population, individuals with two copies of *SMN1* and two copies of *SMN2* are the most common, followed by individuals with two copies of *SMN1* and one copy of *SMN2* [95,96]. These findings suggest that *SMN2* can be lost easily by deletion or gene conversion. Gene conversion events can occur bidirectionally: from *SMN1* to *SMN2* and from *SMN2* to *SMN1*. Ogino et al. presented a piece of evidence of gene conversion from *SMN2* to *SMN1* [97].

As a matter of fact, the mechanisms of copy number variation in the *SMN1* and *2* genes are very complex in humans. Sangaré et al. reported that in sub-Saharan Africa (regions south of the Sahara Desert), the proportion of carriers of SMA (here, individuals with only one copy of *SMN1*) is significantly lower than in North America (Caucasians), Europe, and Asia [98]. They also reported that sub-Saharan Africa had much higher proportions of individuals with three, four or more copies of *SMN1* than North America (Caucasians), Europe, and Asia. Regarding *SMN2*, sub-Saharan Africa also had a very high proportion of individuals with deletion of the *SMN2* gene. Sangaré et al. stated in their paper, “This fact cannot be explained by segmental duplication, gene conversion, or natural selection pressure in malaria, but could be explained by bottleneck effects” [98]. The bottleneck phenomenon in their report means that by chance the population that migrated out of Africa to Asia and Europe had a lower *SMN1* copy number or randomly drifted in this direction after the outmigration. Their explanation is based on the model of “recent African origin of modern humans”.

#### 4.3.4. SMA Phenotype Modifiers Other than *SMN2* Copy Number

As described in the previous Section 2.2.2, Dubowitz described the apparent variation in severity among affected siblings within the same family, suggesting the presence of different grades of severity [15,29]. Jones et al. obtained data from the Cure SMA cohort and reported that among 303 sibships identified from 1996 to 2016, 84.8% were subtype-concordant [99]. However, the rest of the sibling pairs were subtype-discordant. Among the discordant sibships, types II/III and types I/II were the most common pairs. 

Some researchers have carefully studied the phenotypic discordance among the family members in the same SMA family and discovered unexpected phenotype modifiers other than *SMN2* copy number. Oprea et al. reported that PLS3 elevation may be a protective modifier of SMA [100]. Riessland et al. also reported that NCALD suppression may be a protective modifier of SMA [59]. They encountered asymptomatic individuals carrying the same *SMN1* mutations as those carried by their affected siblings, suggesting the influence of modifier genes. 

Oprea et al. discovered that unaffected female patients with *SMN1* deletion exhibited significantly higher expression of the *PLS3* gene than their SMA-affected siblings [100]. The *PLS3* gene product is an F-actin-binding protein that is involved in many cellular processes including endocytosis. In their study using SMA model animals, overexpression of PLS3 protein rescued the axon length and outgrowth defects associated with SMN down-regulation in motor neurons of SMA mouse embryos and in zebrafish. Hao et al. showed that PLS3 protein levels were dependent on SMN protein and that PLS3 protein was able to rescue the neuromuscular defects in SMA [101]. Later, Hosseinibarkooie et al. demonstrated that elevated PLS3 protein levels rescued endocytosis impairment by SMN protein deficit [58].

Riessland et al. also reported that a low level of NCALD protein was associated with a milder phenotype in SMA families [59]. As mentioned above, NCALD protein is a calcium-dependent negative regulator of endocytosis. The beginning of their research also involved analysis of discordant families. To identify the SMA modifier, they combined genome-wide linkage analysis with transcriptome-wide differential expression analysis. A low level of *NCALD* expression was associated with a milder phenotype in the same SMA families. In the analyses of SMA models, *NCALD* knockdown effectively ameliorated SMA-associated pathological defects across species, including worms, zebrafish, and mice. 

Thus, factors other than the *SMN2* copy number, such as PLS3 and NCALD, may create interfamilial discordance in SMA subtypes.

## 5. Drug Development

### 5.1. Development of Therapeutic Options for SMA

As described above, an inverse correlation between the *SMN2* copy number and disease severity suggests that *SMN2* can compensate for the loss of *SMN1* to some extent because *SMN2* can produce a low abundance of full-length SMN protein. Moreover, *SMN2* overexpression rescues embryonic lethality in *Smn* (−/−) mice and results in the birth of mice with SMA [67,68]. Based on these findings, many researchers have come to believe that the most beneficial treatment for patients with SMA is to increase full-length SMN protein in the affected organs, including motor neurons.

In 1995, Lefebvre et al. reported that *SMN1* and *SMN2* genes are related to the pathogenesis of SMA [12], and in 2016, nusinersen became the first drug approved for treatment of SMA in the US [15]. In the 22 years from 1995 to 2016 (Figure 1), a tremendous amount of research was conducted. During this period, researchers spent exciting days working towards a single goal, a cure for SMA, and they created a large and detailed body of knowledge on *SMN1* and *SMN2*, providing hope for the development of therapeutic options for SMA-affected families. Medical advances in this period will be discussed in the following subsections, citing important relevant literatures.

### 5.2. Regulation of SMN2 Splicing

#### 5.2.1. Targeting the Splicing Enhancer in *SMN2* exon 7

In 2003, two artificial compounds that modify the splicing of *SMN2* exon 7 were reported (Figure 5): exon-specific splicing enhancement by small chimeric effectors (ESSENCE) [102] and targeted oligonucleotide enhancers of splicing (TOES) [103]. Both ESSENCE and TOES target ESE in *SMN2* exon 7.

Cartegni et al. found that C-to-T transition in *SMN2* exon 7 disrupts an ESE and that SF2/ASF cannot bind to the ESE, leading to skipping of the index exon [75]. They then designed a new compound, ESSENCE, to regulate *SMN2* exon 7 splicing. The ESSENCE compound had an exon 7-binding peptide nucleic acid (PNA) region and 10 arginine/serine (RS) repeat regions (Figure 5). The ESSENCE compound emulates the ESE-dependent function of serine/arginine rich proteins (SR proteins). SR proteins (including SF2/ASF) bind to an ESE through their RNA-binding domain and promote exon inclusion by recruiting the splicing factors through their RS domain. Using in vitro transcription and splicing systems, the authors demonstrated that the addition of ESSENCE rescued the splicing *SMN2* exon 7 to the *SMN1* exon 7 level [102].

Scordis et al. invented the TOES compound consisting of two oligonucleotides: an antisense oligonucleotide (ASO) that binds to *SMN2* exon 7 and a GGA-repeat oligonucleotide that binds to SF2/ASF [103] (Figure 5). Using in vitro systems and fibroblasts, they also demonstrated that the TOES compound rescued the splicing of *SMN2* exon 7 to the level of *SMN1* exon 7 splicing [103].

#### 5.2.2. Targeting the Splicing Silencers in Flanking Introns of *SMN2* exon 7

Around the same time that the ESSENCE and TOES compounds were reported, different approaches outside *SMN2* exon 7 were also reported. The suppression of intronic splicing silencer (ISS) prevented the exclusion (skipping) of the exon (Figure 5).

In 2002, Miyajima et al. identified a cis-acting element (element 1) that regulated *SMN2* exon 7 splicing in intron 6 [104]. Their minigene experiments demonstrated that a mutation in element 1 and an ASO directed toward element 1 caused an increase in *SMN2* exon 7 inclusion. In 2006, Singh et al. identified a cis-acting element (ISS-N1) that regulated *SMN2* exon 7 splicing in intron 7 [105] (Figure 5). Deletion of ISS-N1 and ASO toward ISS-N1 promoted exon 7 inclusion in mRNAs derived from the *SMN2* minigene.

In 2008, Hua et al. reported a systematic screening of splicing regulatory elements in the flanking intronsAls [106]. To identify potential ISSs that inhibit *SMN2* exon 7 inclusion, they systematically screened the intronic sequences on either side of exon 7. They tested the splicing pattern (exclusion or inclusion of *SMN2* exon 7) in vitro and in cells using 10 ASOs targeting each flanking intronic region. They also identified two ISSs: one in intron 6 and a recently described one in intron 7.

In animal studies, an ISS-N1-targeting ASO drug stimulated the inclusion of exon 7 in *SMN2* mRNA transcripts, resulting in increased production of the full-length SMN with significant amelioration of the survival period and pathological changes in different experimental SMA models [107,108]. Nusinersen is an ASO drug targeting ISS-N1 and will be discussed later in this review article.

### 5.3. Introduction of Exogenous Genes

#### 5.3.1. Lentiviral Gene Transfer System (Lentivector)

In 2004, Azzouz et al. constructed lentivector-LacZ (a lentivector carrying the β-galactosidase gene) and attempted to deliver it to motor neurons via retrograde axonal transport. After multiple injections of lentivector-LacZ into the muscles of mice, the motor neurons were stained blue with the X-Gal reaction [109]. This finding indicated that gene transfer to motor neurons is possible using retrograde axonal transport. The authors then constructed lentivector-SMN (a lentivector carrying *SMN* cDNA) and injected it into the muscles of SMA model mice on postnatal day 2. These procedures reduced motor neuron death and prolonged the average survival time of the SMA model mice. The findings of this study suggest that the exogenous *SMN* gene functioned in the SMA model mice.

#### 5.3.2. Adeno-Associated Virus (AAV) Vector

In 2009, Foust et al. reported that an intravenously injected self-complementary AAV9 (scAAV9) vector crossed the blood–brain barrier and entered cells of the central nervous system [110]. The AAV9 vector targeted cells within the central nervous system. The concept of scAAV will be explained later in this review. When the scAAV9 vector carrying green fluorescent protein (scAAV9-GFP) was injected into neonatal mice through the facial vein on postnatal day 1, GFP was expressed in the cerebrum, cerebellum, and spinal cord.

The next year, the authors reported that the scAAV9 vector carrying the *SMN* gene (scAAV9-SMN) was injected into the facial vein of neonatal SMA model mice on the first day after birth, resulting in prolonged survival of one of the treated mice [111]. (Regrettably, however, their paper has been retracted because of its inaccurate description.)

Onasemnogene abeparvovec is a drug based on scAAV9-SMN and will be discussed later in this review article.

### 5.4. Therapeutic Strategies with Existing Medication

#### 5.4.1. Aclarubicin

With the development of drug repositioning (redevelopment of existing drugs), exploratory studies were repeated to find drugs that would increase SMN protein in patients with SMA. Here, we focus on aclarubicin, valproic acid (VPA), and salbutamol. Many other agents (riluzole, thyroid-stimulating hormone-releasing hormone (TRH), gabapentin, hydroxyurea, etc.) have been tested for the treatment of SMA [112], but will not be included in this review.

In 2001, Andreassi et al. showed that aclarubicin, an anthracycline anticancer drug, blocked *SMN2* gene exon 7 skipping in fibroblasts from patients with SMA [113]. In practice, it is impossible to use a highly toxic anticancer drug such as aclarubicin as a therapeutic agent for SMA. However, the research on aclarubicin is very valuable because it shows that splicing can be modified by a drug with a certain chemical structure. Because aclarubicin is a tetracycline derivative, the above-mentioned report on its splicing correction effect provided the impetus for investigating the tetracycline scaffold as a platform for screening *SMN2* exon 7 activators.

In 2009, Hastings et al. reported that a tetracycline-like antibiotic, PTK-SMA1, blocked exon 7 skipping of *SMN2* in fibroblasts from patients with SMA and in liver from SMA-affected mice [114]. Researchers expected that the mechanism of compounds that correct *SMN2* splicing, such as PTK-SMA1, may be associated with positive or negative regulators of splicing, including SR proteins [115].

After the discovery of compounds that correct *SMN2* splicing, researchers performed high-throughput screening of compounds with similar functions, resulting in the discovery of SMN-C3 [116]. SMN-C3 may be considered the prototype of risdiplam.

#### 5.4.2. Valproic Acid (VPA)

VPA is a well-established anticonvulsant for treatment of several types of seizures, and it works as a histone deacetylase (HDAC) inhibitor. In 2003, two research groups (Brichta et al. in Germany and Sumner et al. in the US) reported that VPA increases full-length SMN protein in fibroblasts derived from patients with SMA [117,118]. Both groups also demonstrated that VPA increased transcription of *SMN2* and corrected splicing of *SMN2* exon 7. VPA modified the splicing pattern of *SMN2* exon 7, probably by altering the expression of genes that produce splicing-related proteins.

In 2006, Weihl et al. reported their experience of treating 7 adult patients with SMA type III/IV with VPA for an average of 8 months [119]. Their VPA-treated patients showed increased quantitative muscle strength and subjective function. In the same year, Brichta et al. reported that 7 of 20 VPA-treated patients showed an increase in the full-length *SMN2* transcript of their peripheral blood cells [120]. Additionally, the authors reported that there were responders and non-responders to VPA treatment. In 2009, Swoboda et al. evaluated 27 patients with SMA type II/III who were treated with VPA for 12 months. VPA improved the scores of the modified version of the Hammersmith Functional Motor Scale for SMA, but the improvements were observed primarily in children aged < 5 years [121].

Although these studies involving a small number of patients with SMA have shown the efficacy of VPA, two large clinical studies with VPA-treated and -untreated groups concluded that VPA was ineffective in improving motor function in patients with SMA. The CARNIVAL TRIAL (I) of patients who were able to sit up but unable to stand or walk did not reach the conclusion that VPA improved clinical symptoms [122]. The CARNIVAL TRIAL (II) of patients who were able to stand and walk also failed to conclude that VPA ameliorated clinical symptoms [123].

Several questions remain about these large-scale studies. One of them is whether there was sufficient stratification in these studies. Stratification of patients in clinical trials is important. According to Garbes et al., one third of patients with SMA are responders to VPA, and a lack of response to VPA is closely associated with increased expression of the fatty acid translocase CD36 [124]. Before enrollment of a patient in a clinical trial, it may be better to perform in vitro studies to predict the patient’s response to the compound under investigation [125].

#### 5.4.3. Salbutamol

Salbutamol is a β-adrenergic agonist bronchodilator that is often used for bronchial asthma. Beta-adrenergic agonists have been demonstrated to increase muscle strength in healthy volunteers and have been used in patients with neuromuscular weakness [126,127].

In 2002, Kinali showed that these effects of β-adrenergic agonists were also seen in patients with SMA. In their study, 13 patients (5 with SMA type II and 8 with SMA type III) were given oral salbutamol for 6 months. Significant increases in muscle strength, forced vital capacity, and lean body mass were noted between the baseline and the 6-month assessments (*p* < 0.05) [128]. In 2008, Pane et al. reported that 1 year of administration of salbutamol (2 mg, 3 times a day) to 23 patients with SMA type II resulted in improvement in the Hammersmith Motor Functional Scale score [129]. In 2019, Tizziano also reported that 1 year of administration of salbutamol (4 mg, 3 times a day) or placebo to 45 adult patients with SMA type III produced significantly better results in motor function in the active drug group than in the placebo drug group [130].

Regarding the mechanism of action of salbutamol in patients with SMA, Angelozzi et al. reported in 2007 that salbutamol promoted transcription of *SMN2* and splicing of *SMN2* exon 7 in fibroblasts derived from a patient with SMA type I, leading to an increase in intracellular SMN protein [131]. However, in 2015, Harahap et al. suggested a different mechanism underlying the increase in SMN protein based on their experiment using fibroblasts derived from a patient with SMA type III. According to Harahap et al., salbutamol may inhibit ubiquitin-mediated protein degradation, resulting in an increase in SMN protein [132]. Taken together, these findings indicate that salbutamol may be a beneficial drug for patients with SMA.

## 6. FDA-Approved Drugs for SMA

### 6.1. Nusinersen

#### 6.1.1. Outline

Historically, no drugs were effective enough to stop the progression of SMA. Thus, SMA was recognized as an incurable disease. However, since the appearance of nusinersen at the beginning of the 21st century, the medical situation around patients with SMA has completely changed. Nusinersen was approved in the US in 2016 (Table 2) and in Europe and Japan in 2017.

Nusinersen (Spinraza^®^; Biogen, Cambridge, MA, USA) is an ASO drug targeting an ISS site in *SMN2* intron 7, ISS-N1. For delivery of drugs to motor neurons in the central nervous system, direct injection into the cerebrospinal fluid within the intrathecal space may be necessary because ASOs do not cross the intact blood–brain barrier. Thus, nusinersen is injected intrathecally three or four times over two months during the initial loading period and every four or six months during the maintenance period [133].

#### 6.1.2. Mechanism of Action

Nusinersen binds to ISS-N1 (or masks ISS-N1) and inhibits the binding of other splicing factors, stimulating the inclusion of exon 7 into mRNA. Hua et al. proposed that the inhibitory effect of ISS-N1 is due to the interaction of ISS-N1 with hnRNP A1/A2 [106] (Figure 6).

However, Singh et al. suggested that ISS-N1 may be a composite landing site for several splicing factors. They did not believe that hnRNP A1/A2 was the sole regulator associated with the negative effect of ISS-N1. Instead, they proposed a different mechanism, alteration of the secondary structure of pre-mRNA, by which the ISS-N1-targeting ASO promotes the recruitment of U1 snRNP at the 5′-splice site of exon 7 [134] (Figure 6). U1 snRNP binds to the 5′ exon–intron junction of pre-mRNA and thus plays a crucial role at an early stage of pre-mRNA splicing.

#### 6.1.3. Clinical Trials

##### Clinical Trials Involving Symptomatic Patients

The ENDEAR trial enrolled 122 infants with SMA types I and II, two thirds (*n* = 81) of whom were treated with nusinersen and one third (*n* = 41) of whom received sham treatment [135]. The infants had an *SMN1* deletion or mutation, had two copies of *SMN2*, and developed symptoms by 6 months of age. The primary endpoints were the proportion of motor milestone responders (evaluated by Hammersmith Infant Neurological Examination Section 2 (HINE2)) and event-free survival (evaluated by the time to death or use of permanent ventilatory support). The secondary endpoints included subgroup analyses of the overall survival and event-free survival rates according to the disease duration at screening. The last evaluation was performed on day 394 after treatment initiation. In the final analysis, a significantly higher proportion of infants in the nusinersen than the control group had a motor milestone response (37 of 73 [51%] vs. 0 of 37 [0%] infants, respectively). The event-free survival rate was significantly higher in the nusinersen group than in the control group. The overall survival rate was also significantly higher in the nusinersen group than in the control group. Notably, infants with a shorter disease duration at screening were more likely to benefit from nusinersen than those with longer disease duration.

The CHERISH trial enrolled 126 children with SMA types II and III, two thirds (*n* = 84) of whom were treated with nusinersen and one third (*n* = 42) of whom received sham treatment [136]. The children had deletions or mutations in *SMN1* and two or more copies of *SMN2*. Most of them carried three copies of *SMN2*. They developed symptoms after 6 months of age and had the following characteristics at screening: age of 2 to 12 years and the ability to sit independently but no history of walking independently (defined as the ability to walk ≥15 feet unaided). The primary endpoint was the least-squares mean change from baseline in the Hammersmith Expanded Functional Motor Scale (HFMSE) score at 15 months after treatment initiation. The secondary endpoint was the percentage of children with a clinically meaningful increase from baseline in the HFMSE score (≥3 points indicated improvement in at least two motor skills). At the interim analysis, the nusinersen group showed a least-squares mean increase of 4.0 points in the HFMSE score from baseline to month 15, whereas the control group showed a least-squares mean decrease (−1.9 points). This between-group difference was significant and in favor of nusinersen (least-squares mean difference in change, 5.9 points; 95% confidence interval, 3.7–8.1; *p* < 0.001). At the final analysis, 57% of children in the nusinersen group showed an increase in the HFMSE score of at least 3 points from baseline to month 15, compared with only 26% of children in the control group (*p* < 0.001).

##### Clinical Trials Involving Pre-Symptomatic Patients

The NURTURE trial enrolled 25 children with genetically diagnosed SMA who had deletions or mutations in *SMN1* and two or three copies of *SMN2* [137]. The primary endpoint for NURTURE was time to death or respiratory intervention (invasive or non-invasive for ≥6 h per day continuously for ≥7 days or tracheostomy). The secondary endpoint included proportion of participants alive, attainment of motor milestones, maintenance of weight, etc. Motor milestones were assessed by scores on the World Health Organization (WHO) criteria, HINE-2, and the Children’s Hospital of Philadelphia Infant Test of Neuromuscular Disorders (CHOP INTEND) motor function scale.

The participants were first administered nusinersen in infancy while still pre-symptomatic. When the report of the trial was published, the children were already past the expected age of symptom onset for SMA type I or II. At the time, all participants were alive and none required constant ventilation. Additionally, all had achieved the ability to sit without support, 23 (92%) were able to walk with assistance, and 22 (88%) were able to walk independently. These results may underscore the importance of proactive treatment with nusinersen immediately after genetic diagnosis of SMA in pre-symptomatic infancy, suggesting the effectiveness of newborn screening for SMA.

### 6.2. Onasemnogene Abeparvovec

#### 6.2.1. Outline

Onasemnogene abeparvovec is the next new drug that was introduced into clinical practice after nusinersen. The drug was developed for *SMN* gene (more precisely, full-length *SMN* cDNA) transfer. It was approved in the US in 2019 (Table 2) and in Europe and Japan in 2020.

Govoni et al. stated, “SMA is considered an optimal candidate disease for gene therapy for several reasons. First, without treatment, SMA would be a fatal disease. Second, the specific genetic target (*SMN1*) and cell population (lower motor neurons) affected in SMA are known. Third, toxicity from gene therapy would be minimal because SMN protein overexpression in tissues other than lower motor neurons is well tolerated and the presence of *SMN2* reduces the risk of an autoimmune response. Fourth, preclinical studies can be developed owing to robust animal models for SMA and *SMN* cDNA length that are appropriate for gene therapy vectors” [138].

Onasemnogene abeparvovec (Zolgensma^®^; Novartis Gene Therapies, Bannockburn, IL, USA) is a drug used for patients with all forms and types of SMA who are <2 years of age at the time of dosing. This drug is a one-time treatment given through an intravenous infusion that takes 1 h [139].

#### 6.2.2. Mechanism of Action

Onasemnogene abeparvovec is an scAAV9 vector-based drug crossing the blood–brain barrier. AAV vectors do not integrate into host DNA. After entering the host cell, the AAV vector translocates into the nucleus, where the transgene acts as an episome (a small, stable chromosome separate from the native chromosome) (Figure 7). However, the gene-expression efficiency of AAV vectors packaging single-stranded DNA is not high. This is because double-stranded DNA must be synthesized prior to gene expression. 

In onasemnogene abeparvovec, to increase the efficiency of gene expression, the scAAV vector packages the double-stranded genome [140]. In addition, the scAAV ITR increases the speed at which the double-stranded transgene is transcribed and the resulting protein is produced. Hybrid CMV enhancer and CB promoter activates the transgene to allow for continuous and sustained SMN protein expression.

#### 6.2.3. Clinical Trials

##### Clinical Trials Involving Symptomatic Patients

The START trial enrolled 15 infants with SMA type I who had deletions in *SMN1* and two copies of *SMN2* [141]. The infants were divided into two onasemnogene abeparvovec treatment cohorts: a low-dose cohort (6.7 × 10^13^ vg/kg, *n* = 3) and a high-dose cohort (2.0 × 10^14^ vg/kg, *n* = 12). The mean age of the infants in the low-dose cohort was 6.3 months (range, 5.9–7.2 months), and that of the infants in the high-dose cohort was 3.4 months (range, 0.9–7.9 months). The primary outcome of this clinical trial was safety. The secondary outcome was the time until death or the need for permanent ventilatory assistance. In the exploratory analyses, the scores on the CHOP INTEND scale of motor function were compared between the two cohorts, and motor milestones in the high-dose cohort were also compared with scores in studies of the natural history of the disease (historical cohorts).

The high-dose cohort showed a rapid increase from baseline in the CHOP INTEND score following gene delivery (increase of 9.8 points at 1 month and 15.4 points at 3 months). By contrast, this score decreased in a historical cohort. Of the 12 patients who had received the high-dose treatment, 11 sat unassisted, 9 rolled over, 11 fed orally and could speak, and 2 walked independently.

The 5-year extension results of the START trial have already been reported [142]. The report indicates that 10 patients in the high-dose cohort (2 patients’ families declined follow-up participation) were alive and did not require permanent ventilation. In addition, the motor milestones achieved in the START study have been maintained throughout long-term follow-up; furthermore, 2 patients attained the new milestone of standing with assistance.

The START trial showed significant improvement in the motor function of patients in the high-dose cohort who were treated before 3 months of age (early dosing) and had high motor function (CHOP INTEND score of ≥20 points) at the time of treatment. By contrast, there was little improvement in the motor function of patients who were treated after 3 months of age or had low motor function (CHOP INTEND score of <20 points) at the time of treatment, even if they were in the high-dose cohort [141,142]. Thus, two factors may determine the prognosis for motor function: the timing of treatment (early or late) and the motor function at the time of treatment (high or low) [143].

##### Clinical Trials Involving Pre-Symptomatic Patients

The SPR1NT trial enrolled 29 infants with genetically diagnosed SMA who had deletions or mutations in *SMN1* and two or three copies of *SMN2* [144,145]. They were treated with onasemnogene abeparvovec in infancy while still pre-symptomatic. The participants were divided into 2 cohorts based on their *SMN2* copy number: Cohort 1 (14 infants with 2 copies of *SMN2* expected to develop SMA) and Cohort 2 (15 infants with 3 copies of *SMN2* expected to develop SMA). Efficacy was compared with a matched natural history cohort (23 infants).

In Cohort 1 [144], the primary endpoint was the ability to sit independently for ≥30 seconds at any visit up to 18 months of age. The first secondary efficacy endpoint was the percentage of patients who survived and did not require permanent ventilation at 14 months of age, and the second secondary endpoint was maintenance of weight at or above the third World Health Organization percentile without feeding support at any visit up to 18 months of age. The exploratory endpoints were achievement of motor milestones as assessed by the World Health Organization Multicentre Growth Reference Study (WHO-MGRS) and Bayley Scales of Infant Development version 3 gross motor criteria, CHOP INTEND scores, and scores on the Bayley Scales of Infant Development gross and fine motor subtests. In this cohort (Cohort 1) [144], all 14 children with 2 *SMN2* copies treated at median age of 21 days of life (range, 8–34 days) sat independently for ≥30 seconds at any visit at ≤18 months of age (11 within the normal developmental window). All survived without permanent ventilation at 14 months, and 13 maintained their body weight without feeding support through 18 months. None required nutritional or respiratory support. No serious adverse events were considered treatment-related by the investigator.

In Cohort 2 [145], the primary efficacy endpoint was the ability to stand independently for ≥3 seconds at any visit up to 18 months of age. The secondary efficacy endpoint was the ability to walk alone for at least five steps at any visit up to 24 months of age. The exploratory endpoints were survival at 14 months of age (defined as the avoidance of death or a requirement for permanent ventilation) and the ability to maintain body weight at or above the third percentile without the need for feeding support at any visit up to 24 months of age. In this cohort (Cohort 2) [145], all 15 children with 3 *SMN2* copies treated before symptom onset stood independently before 24 months of age (14 within the normal developmental window), and 14 walked independently (11 within the normal developmental window). All survived without the need for permanent ventilation at 14 months, and 10 maintained their body weight without feeding support through 24 months. None required nutritional or respiratory support. No serious adverse events were considered treatment-related by the investigator.

Based on the clinical trial results of the two cohorts, onasemnogene abeparvovec was effective for and well tolerated by pre-symptomatic infants at risk of SMA type I or II, highlighting the urgency for early identification (newborn screening) and intervention [144,145]. However, serious adverse events must also be mentioned. In the post-marketing setting, transient hepatotoxicity (including four cases of acute liver failure) was the most common adverse event. Thrombotic microangiopathy was also observed in the post-marketing setting [144,145].

### 6.3. Risdiplam

#### 6.3.1. Outline

Risdiplam (Evrysdi^®^; Roche, Basel, Switzerland) is the third drug introduced into clinical practice after nusinersen and onasemnogene abeparvovec, but it is the first FDA-approved prescription medicine for SMA. It was approved in the US in 2020 (Table 2) and in Europe and Japan in 2021. Originally approved for the treatment of patients with SMA who are >2 months of age, risdiplam is now approved in the US for pediatric and adult patients with SMA of all ages.

Risdiplam is an oral solution containing a small-molecule compound that modifies *SMN2* pre-mRNA splicing. It allows patients with SMA to receive treatment without leaving their homes. This medicine is given once a day; dosing is dependent on age and body weight. Adults and children aged > 2 years and weighing ≥ 20 kg take 5 mg of risdiplam once a day. For children ≥ 2 years of age weighing < 20 kg, the dose is based on body weight (usually 0.25 mg/kg once a day) [146].

#### 6.3.2. Mechanism of Action

Risdiplam is distributed in central and peripheral cells and acts in the nucleus. It regulates the alternative splicing of *SMN2*. However, there is no consensus on the mechanism by which risdiplam promotes *SMN2* exon 7 inclusion with high specificity [147].

Previous studies have suggested that it may act on two regions of *SMN2* exon 7 for highly selective pre-mRNA splicing (Figure 8) [147,148]. In *SMN2* pre-mRNA, exon 7 and its flanking regions have two stem-loop structures (terminal stem-loop 1 (TSL1) near 3′-splice site and terminal stem-loop 2 (TSL2) near 5′-splice site) and they are closely related to alternative splicing of *SMN2* exon 7 [149]. Risdiplam may influence these stem-loop structures.

The first region is the 5′-splice site of exon 7 of the pre-mRNA transcribed from the *SMN2* gene. A risdiplam analogue may bind to the stem-loop structure, TSL2, at the end of *SMN2* exon 7 and stabilize the duplex of the 5′-splice site RNA sequence and the U1 snRNP RNA sequence and promote splicing initiation [147,150].

The second region is the internal structure around exonic splicing enhancer 2 (ESE2) at the middle of *SMN2* exon 7. A risdiplam analogue binds to the ESE2 region, which alters the stem-loop structure, TSL1, in the first half of *SMN2* exon 7. Risdiplam analogues bound to the ESE2 region inhibit hnRNPG protein binding while promoting the binding of two splicing regulatory proteins: far upstream element binding protein 1 (FUBP1) and its homolog, KH-type splicing regulatory protein (KHSRP) [150,151]. TSL1 includes a cis-acting element that suppresses *SMN2* exon 7 splicing, and FUBP1 and KHSRP proteins are trans-acting factors that promote *SMN2* exon 7 splicing.

#### 6.3.3. Clinical Trials

##### Clinical Trials Involving Symptomatic Patients 

Four clinical trials have been performed: the FIREFISH trial for patients with SMA type I, the SUNFISH trial for patients with SMA type II/III, the JEWELFISH trial for patients with SMA who underwent some treatment previously, and the RAINBOWFISH trial for pre-symptomatic patients. We herein explain the main points of three of these trials: FIREFISH, SUNFISH, and RAIBOWFISH.

The FIREFISH trial (Part 1) enrolled 21 infants with SMA type I [152]. The participants were divided into 2 risdiplam treatment cohorts: the low-dose and high-dose cohorts. The low-dose cohort comprised 4 infants who were treated with a final dose of 0.08 mg/kg per day at month 12, and the high-dose cohort comprised 17 infants who were treated with a final dose of 0.2 mg/kg per day at month 12. Risdiplam treatment increased the median blood concentration of SMN protein in the infants of both cohorts. In total, 7 infants in the high-dose cohort and no infants in the low-dose cohort were able to sit without support for at least 5 seconds. Thus, the higher dose of risdiplam (0.2 mg/kg per day) was selected for part 2 of the study. 

The FIREFISH trial (Part 2) enrolled 41 infants with SMA type I, aged 1 to 7 months at enrolment, with a genetically confirmed diagnosis of SMA and two *SMN2* gene copies [153]. The primary endpoint was the ability to sit without support for at least 5 seconds after 12 months of treatment. Key secondary endpoints were a score of 40 or higher on the CHOP-INTEND score, an increase of at least 4 points from baseline in the CHOP-INTEND score, HINE-2 score, and survival without permanent ventilation. For the secondary endpoints, comparisons were made with the upper boundary of 90% confidence intervals for natural-history data from 40 infants with type 1 SMA. After 12 months of treatment, 12 infants (29%) were able to sit without support for at least 5 seconds, a milestone not attained in this disorder [153]. After 24 months of treatment, 18 infants (44%) were able to sit without support for at least 30 seconds. However, no infants could stand alone after 24 months of treatment [154].

The SUNFISH trial (Part 1) was an exploratory dose-finding study involving 51 patients with SMA type II or III, and it was performed to determine the dose for use in part 2 of the study [155]. The SUNFISH trial (Part 2) involved 180 patients with SMA type II and non-ambulant type III aged 2 to 25 years of age [156]. The primary endpoint was the change from baseline in the 32-item Motor Function Measure (MFM32) total scores at 12 months of treatment. The secondary endpoints were the proportion of patients who had marked improvement (a change from baseline of ≥3 points) in MFM32, changes from baseline in the scores of Revised Upper Limb Module (RULM) and Spinal Muscular Atrophy Independence Scale (SMAIS; reported by caregivers).

In the SUNFISH trial (Part 2), the patients were stratified by age and randomly assigned (2:1) to receive either daily oral risdiplam (5.00 mg for individuals weighing ≥20 kg and 0.25 mg/kg for individuals weighing < 20 kg) or a daily oral placebo. After 1 year of risdiplam treatment, the least-squares mean change from baseline in the MFM32 total scores was 1.36 in the risdiplam group and −0.19 in the placebo group, with a treatment difference of 1.55 in favor of risdiplam.

A one-year follow-up of the SUNFISH trial (Part 2) showed that the safety profile after 24 months of treatment was consistent with that observed at 12 months of treatment [157]. Risdiplam over 24 months resulted in further improvement or stabilization in motor function, confirming the benefit of longer-term treatment. The subgroup analyses showed that motor function was generally improved in younger individuals and stabilized in older individuals [157].

##### Clinical Trials Involving Pre-Symptomatic Patients

The RAINBOWFISH trial is an ongoing study assessing the efficacy, safety, and pharmacokinetics/pharmacodynamics of risdiplam. The study includes infants with genetically diagnosed pre-symptomatic SMA from birth to 6 weeks of age (at first dose), regardless of the *SMN2* copy number [158]. The primary analysis will be conducted at month 12 in infants with two *SMN2* copies and a baseline compound muscle action potential amplitude of ≥1.5 mV. The primary endpoint is the proportion of infants sitting without support for ≥5 seconds. The secondary endpoints are the clinical manifestation of SMA, survival and permanent ventilation, motor milestone achievement, motor function, growth measures, nutritional status, compound muscle action potential, pharmacokinetics/pharmacodynamics, and safety monitoring.

According to the interim report [158], the median age at the first dose was 26.5 days (range, 16–40 days) for the first 18 enrolled infants (data cut-off, 1 July 2021). A total of 7 infants have been treated for ≥12 months, and the preliminary efficacy data demonstrate that most reached near-maximum score organization windows for healthy children. All infants treated for ≥12 months remain alive without permanent ventilation, have maintained their swallowing and feeding abilities, and have not required hospitalization.

## 7. Necessity of SMA Newborn Screening

### 7.1. Detection of Pre-Symptomatic SMA

With the introduction of effective therapies with new drugs (nusinersen, onasemnogene abeparvovec, and risdiplam) into the clinical setting, the timing and accuracy of SMA diagnosis have become much more important than ever before. If the treatment cannot be initiated at the proper time because of delayed diagnosis or misdiagnosis, the therapeutic efficacy may be markedly reduced.

Clinical trials of nusinersen and onasemnogene in pre-symptomatic patients showed that nearly all patients achieved good motor function. Patients enrolled in these clinical trials were those who, without treatment, would be expected to exhibit symptoms of SMA type I or II. If treated before symptoms develop, even infants predicted to have SMA type I (genetically diagnosed but asymptomatic) would be able to sit without support [137,144]. Additionally, if treated before symptoms develop, even infants predicted to have SMA type II (genetically diagnosed but asymptomatic) would be able to walk independently [137,145].

Following the success of clinical trials, population-based newborn screening programs for SMA (more precisely, *SMN1*-deleted SMA) have been increasingly implemented worldwide [159]. As of 29 December 2020, neonatal screening programs for SMA were available in Taiwan, USA, Germany, Belgium, Australia, Italy, Russia, Canada, and Japan [160].

Establishing a neonatal screening system for SMA is essential to initiate treatment in the pre-symptomatic stage and to maximize the therapeutic efficacy.

It should be noted here that neonatal screening for SMA detects homozygous deletion of the *SMN1* gene, not intragenic *SMN1* mutations.

### 7.2. Prevention of Delayed Diagnosis of SMA

In the following discussion, we assume that the disease had already developed before the results of newborn screening were known and that pre-symptomatic treatment was impossible. Even in such cases, newborn screening would still be useful for patients with SMA.

It has long been noted that the diagnosis of SMA tends to be delayed [161]. Before SMA newborn screening started, the diagnosis of SMA type I was often delayed until the sixth month of life [161]. SMA types II and III were diagnosed at 20 and 50 months of age, respectively [161].

However, newborn screening would prevent such delays in diagnosis and allow patients to receive treatment as early as possible. Both pre-symptomatic and symptomatic SMA can now be detected by newborn screening and treated within 3 to 4 weeks after birth.

Kariyawasam et al. reported that patients who underwent SMA newborn screening achieved higher levels of motor function than those who did not undergo newborn screening [162]. They reported the outcomes of therapeutic treatment for symptomatic infants with SMA identified by newborn screening (most infants with symptomatic SMA carried two *SMN2* copies). According to their report, newborn screening enabled infants with symptomatic SMA to access therapeutic treatment much earlier than before, resulting in incredibly good outcomes. Kariyawasam et al. demonstrated that with newborn screening followed by early treatment, even patients with *SMN1* deletion who have two *SMN2* copies could stand with/without assistance or walk with assistance.

In addition, SMA newborn screening is very useful for patients with SMA who have comorbidities. Differential diagnosis for hypotonia is difficult, particularly in the neonatal period. Development of respiratory or other problems in an infant with hypotonia can hamper the suspicion of SMA [163]. In such cases, SMA cannot be diagnosed without newborn screening, and its diagnosis will be delayed by several weeks or months because of the time needed for clinical referrals and investigations.

### 7.3. Treatment Algorithm for Patients Identified by Newborn Screening

In 2018, Glascock et al. published a report titled “Treatment Algorithm for Infants with SMA Detected by Newborn Screening” [164]. Newborn screening for SMA detects pre-symptomatic and symptomatic patients. Therefore, this treatment algorithm is not based on their symptoms but instead on their *SMN2* copy number.

If patients carry only one *SMN2* copy and show any symptoms (probable SMA type 0), initiation of treatment is left to the discretion of the physicians. If patients have two *SMN2* copies (probable SMA type I), early initiation of treatment is required because symptoms of SMA type I are expected to develop without treatment. If three *SMN2* copies are present (probable SMA type II or III), early initiation of treatment is required because symptoms of SMA type II or III are expected to develop without treatment.

If patients carry four or more copies (probable SMA type III or IV), early initiation of treatment may not be necessary, but the progress should be monitored carefully and treatment initiated later in life, when signs of SMA development are observed.

In 2020, however, the same group published a revised version of the above algorithm [165]. According to the revised version, even if the *SMN2* copy number is four or more, the patient should be treated as soon as possible. With early treatment, the disease would be mostly eradicated in pre-symptomatic patients with four *SMN2* copies. In addition, when the *SMN2* copy number is four or more, the copy number is difficult to accurately measure (methodological problem in screening).

In an SMA newborn screening pilot project in Germany conducted by Müller-Felber et al., 278,970 infants were screened from January 2018 to November 2019, and 38 positive cases with a homozygous *SMN1* deletion were detected. Of these cases, 40% had four or more *SMN2* copies [166]. The incidence of homozygous *SMN1* deletion was 1:7350. Of the 15 children with SMA who had 4 *SMN2* copies, 1 child developed physical signs of SMA by the age of 8 months. Reanalysis of the *SMN2* copy number using a different test method revealed three copies.

According to the follow-up to the above report [167], two families with newborns with four *SMN2* copies reported during follow-up that the respective 5-year-old and 6-year-old brothers had unclear motor symptoms; both of them showed some walking disturbance at 3 years of age. In the two screened index patients, the start of treatment within the first year of life irrespective of the clinical status is under discussion with the parents. At present, immediate treatment might be recommended for patients with SMA detected during newborn screening, regardless of their *SMN2* copy number.

### 7.4. Obstacles to Implementation of Newborn Screening for SMA

As mentioned above, effective therapies with new drugs (nusinersen, onasemnogene abeparvovec, and risdiplam) have been introduced in the clinical setting. However, this is only practiced in a limited number of countries worldwide [160]. Despite the appearance of these new drugs, many countries have neither introduced new drugs, nor implemented newborn screening for SMA.

In many countries, the high cost of the new drugs (nusinersen, onasemnogene abeparvovec, and risdiplam) hampers introduction of new treatments for SMA and/or implementation of newborn screening for SMA. People believe that newborn screening for SMA needs to be coupled with access to treatments with expensive drugs. To establish a national consensus on implementation of newborn screening for SMA, accurate and persuasive cost–benefit data of SMA newborn screening followed by treatment with new drugs should be collected in individual countries [160].

However, the high cost of drugs is not the only obstacle to the implementation of newborn screening for SMA. Low public awareness of SMA and stigmatization against SMA may also be major obstacles [168]. A recent study in Japan, where newborn screening for SMA is not yet widespread, found that many Japanese people have very limited knowledge of SMA and its new treatments [169]. The result suggested that raising public awareness of SMA should be the first step towards establishing a national consensus on implementing newborn screening for SMA in each country. In addition, along with raising public awareness of SMA, social issues regarding stigmatization or genetic discrimination against SMA patients or SMA-screen-positive infants and their families should be overcome in any country. 

## 8. Discussion

### 8.1. Four Historical Mysteries of SMA

There were two unresolved issues regarding the early history of SMA. Firstly, the disease discovered by Werdnig and Hoffmann was SMA type II, but their names are associated with SMA type I (Werdnig–Hoffmann disease). We believe that this is likely to be because Werdnig and Hoffmann attracted attention as the discoverers of the disease causing hypotonia in infancy (then many cases were diagnosed as amyotonia congenita), which was a hot topic in the early 20th century. Secondly, despite its varied clinical manifestations, SMA has come to be regarded as a single disease. This is likely to be because all loci of SMA types I to III are mapped to the same locus, chromosome 5q13.

There were also two ambiguities in the modern history of SMA. Firstly, three drugs with completely different mechanisms of action appeared simultaneously in the early 21st century. We consider that this occurred because all of the three cutting-edge medical technologies (antisense oligonucleotides regulating gene expressions, gene therapy using viral vectors, and high-throughput screening of low-molecular-weight compounds) were applied to drug development for SMA in the latter half of the 20th century. Secondly, the concept of pre-symptomatic treatment based on genetic diagnosis has been rapidly accepted worldwide. This is likely to be because a clinical trial in genetically predisposed newborns showed highly successful results.

The authors will be very pleased if this review succeeds in addressing these four historical mysteries of SMA.

### 8.2. Current Problems Associated with New Therapies

All three drugs described in this review (nusinersen, onasemnogene abeparvovec, and risdiplam) are currently available. These drug treatments could significantly change the disease trajectory from the known natural course of SMA [170,171,172,173].

However, we are now facing problems that did not exist before new treatments for SMA emerged. Parents of infants first diagnosed with SMA worry about which drug to choose. Parents of younger children who have been treated with one of the three drugs begin to worry about whether to change drugs because they are seeing less improvement than expected.

Older children and adults living with SMA, representing two thirds of the overall SMA population, may not have been treated early in their disease course because of lack of treatment availability [174]. Many of these patients are also concerned about whether to be treated with nusinersen or risdiplam (most of them are over two years old and onasemnogene abeparvovec is not available to them.) In addition, the efficacy of new drugs may be very restricted in older children and adults. A report on adult patients treated with nusinersen showed that, although half of the patients felt subjective improvement in function, there were no significant objective changes, which may point largely to a placebo effect [175].

### 8.3. Switching Therapies in Adults with SMA

Since nusinersen was approved in the US in 2016, SMA patients, their families and physicians have been excited about a new era of SMA treatment. In Italy, many unfollowed SMA patients came to the tertiary centers after the introduction of nusinersen [176]. Similarly, in Japan, the detection rate of SMA patients on Shikoku Island increased after the introduction of nusinersen to the clinical sites [177].

The high expectations that patients and families have had for nusinersen have been very difficult to meet, leading to disappointment, frustration, and discontinuation of treatment [178]. Agosto et al. reported a frustrating clinical situation with nusinersen treatment. They experienced some nusinersen-treated patients with adverse events or lack of efficacy. Considering the patients’ anxiety and distress from the burdensome procedure of intrathecal injection over a lifetime, and the frustration with no perceived improvement, they became skeptical about giving all patients nusinersen treatment [179]. They finally provided extensive information about the possibility of switching from nusinersen to risdiplam treatment for the patients [178].

Conversely, there are also reports of switching from risdiplam to nusinersen. Pitarch-Castellano et al. described a patient who changed from nusinersen to risdiplam but then returned to nusinersen because of risdiplam-related leukocytoclastic vasculitis [180]. In the real world, some patients switch from nusinersen to risdiplam, while others switch from risdiplam to nusinersen. 

In the treatment of SMA in adults, switching therapies from nusinersen to risdiplam may be closely related to the limited efficacy and burdensome procedures of nusinersen. Switching therapies from risdiplam to nusinersen may be closely related to the serious adverse effects of risdiplam. However, there are no consensus guidelines on treatment choices or changes in therapy, and treatment decisions are made on a case-by-case basis [181]. Clinicians treating patients with SMA should explain the advantages and disadvantages of each drug to the patients or their parents.

### 8.4. Switching Therapies in Infants with SMA

In the treatment of SMA in neonates and infants, switching between *SMN2*-modifying drugs (nusinersen or risdiplam) and a gene therapy drug (onasemnogene abeparvovec) is a major issue. 

In 2021, Ferrante et al. reported the first case of switching from nusinersen to onasemnogene abeparvovec [182]. They described a preterm infant (male) with SMA type I in whom nusinersen was used as a bridge to gene therapy with onasemnogene abeparvovec. He had been diagnosed as SMA type I by amniocentesis. The parents wanted the baby to be treated with gene therapy after birth, but it could not be done because of the potential adverse effects of steroids on long-term neurodevelopmental outcomes of preterm infants. Steroids are required in the gene therapy with onasemnogene abeparvovec. A joint decision was made between the parents and the treatment team to begin treatment with nusinersen. Such bridging to onasemnogene abeparvovec is almost the same as switching to onasemnogene abeparvovec. If treatment with nusinersen is discontinued after administration of onasemnogene abeparvovec, the therapy of the patient should be considered to be a case of switching from nusinersen to onasemnogene abeparvovec.

In 2022, Tosi et al. reported the first case of switching from risdiplam to onasemnogene abeparvovec [183]. They treated an SMA type I term infant (female) treated with risdiplam by two months and switched to onasemnogene abeparvovec at five months. According to the report, because risdiplam was not available in the family’s home country, her parents wanted her to undergo a once-in-a-lifetime gene therapy treatment with onasemnogen abeparvovec.

### 8.5. Future Prospects for SMA Therapies

In the near future, the standard of care for SMA will be early diagnosis using newborn screening followed by early treatment with new drugs. However, since there are cases in which the new drugs are not sufficiently effective, it is necessary to continue improving treatment methods for SMA.

Harada et al. treated five patients with SMA type 1 with onasemnogene abeparvovec and nusinersen [184]. The regimen of three out of five patients was not a case of switching therapies from nusinersen to onasemnogen abeparvovec. These patients continued to receive treatment with nusinersen after the administration of onasemnogen abeparvovec. Therefore, this regimen should be included in the combination therapy category. Although the safety and efficacy of the combination therapy were shown in these patients, large clinical studies will be needed to confirm the safety and efficacy of combination therapies. 

Two clinical trials of combination therapy for SMA are currently underway. One is the RESPOND study, which is focusing on the combination of two drugs that increase the amount of SMN protein, nusinersen and onasemnogene abeparvovec. The other is the TOPAZ study, which is focusing on the combination of two drugs that have different effects: increasing the amount of SMN protein (nusinersen) and improving muscle function (apitegromab) [173]. Apitegromab is a selective inhibitor of the activation of latent myostatin. We look forward to adding information from real-world clinical experience to the evidence gathered in these clinical trials. 

How about the possibility of combination therapies of *SMN2* modifying drugs (such as nusinersen) and existing drugs (such as VPA)? In 2020, Pagliarini et al. suggested that HDAC inhibitors can potentiate the activity of nusinersen [185]. In 2022, Marasco et al. demonstrated that treating SMA cell and animal models with VPA approximately doubles the rate of transcriptional elongation, significantly improving the efficiency of nusinersen-induced exon 7 inclusion [186]. These findings support the idea that such combination therapies approaches may increase the chances of SMA treatment scenarios.

A new genome-editing-based approach to the treatment of motor neuron diseases, including SMA, has recently emerged [187]. Although this therapeutic strategy may have been validated in cell and animal studies, further studies are needed before it is applied to SMA therapies. Even so, it is certain that the application of these strategies to SMA and other diseases will improve human well-being in future.

### 8.6. Conclusions

The history of SMA research may be divided into three periods. The first period (1891–1994) began with the first report of patients with SMA, and it included classification of clinical subtypes and chromosomal mapping of the SMA locus. The second period (1995–2015) began with cloning of the *SMN* genes, and it included drug repositioning and the development of new drugs for SMA. The third period (2016–present) began with the introduction of FDA-approved drugs for SMA into the clinical setting, and it includes newborn screening for early diagnosis of SMA. Early detection with newborn screening and early treatment with these new drugs will soon become the standard of care for SMA. However, it should be remembered that the benefits of newborn screening and early treatment of SMA are limited to newborns, and that many issues remain unresolved in older children and adults living with SMA. 

Although SMA had long been an incurable disease, the development of new drugs changed the lives of patients with SMA at the beginning of 21st century. Looking back on the history of SMA, we were overwhelmed by the great achievements of our predecessors and inspired by the outstanding researchers of our time. We hope that this article will help young doctors and researchers become more familiar with the history of SMA.

## Figures and Tables

**Figure 1 ijms-24-11939-f001:**
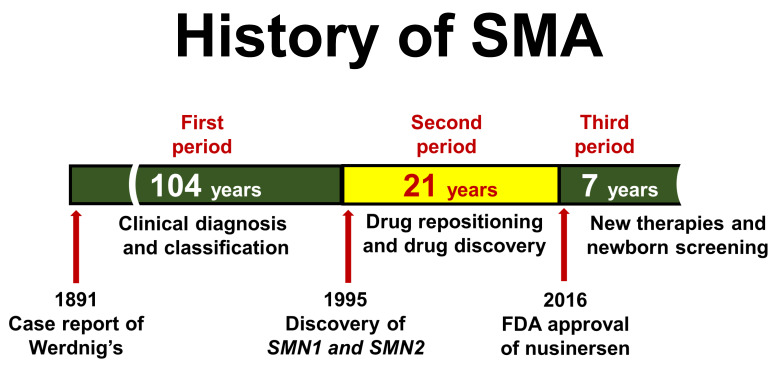
Three periods in the history of SMA research. In 1891, Werdnig reported the first two cases of SMA. Many cases of SMA with different phenotypes have since been reported. In 1995, two SMA-related genes, *SMN1* and *SMN2*, were identified. *SMN1* is the gene responsible for SMA, whereas *SMN2* is a modifier gene of SMA. The discovery of these genes led to the emergence of novel drugs for SMA. In 2016, nusinersen was the first drug approved for treatment of SMA by the FDA in the US [15]. Two other drugs were thereafter approved by the FDA: onasemnogene abeparvovec (approved in 2019) and risdiplam (approved in 2020) [15].

**Figure 2 ijms-24-11939-f002:**
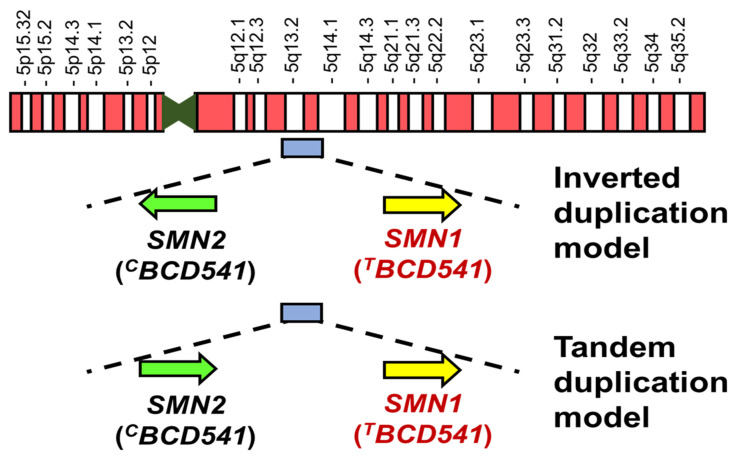
*SMN* genes in chromosome 5q13. In 1995, two SMA-related genes, *SMN1* and *SMN2,* were identified in 5q13 region [12]. These two genes were paralogs. The *SMN1* gene, located in the telomeric side, is the gene responsible for SMA; its loss or defect causes SMA with different phenotypes. The *SMN2* gene, located in the centromeric side, is a modifier gene for SMA; its copy number is associated with the severity of the disease. *SMN1* and *SMN2* were originally reported to exist in an inverted duplication [12]; however, recently, a tandem duplication model has been presented [36].

**Figure 3 ijms-24-11939-f003:**
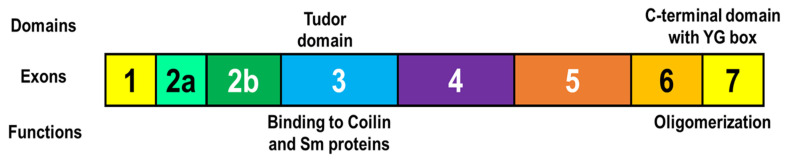
Functional domains of SMN protein. The Tudor domain is responsible for an interaction with coilin, a marker of Cajal bodies [49]. This domain also binds to Sm proteins [50,51]. The YG box is a tyrosine/glycine-rich region in the C-terminus of SMN protein that facilitates oligomerization of SMN protein by formation of the glycine zipper structure [52].

**Figure 4 ijms-24-11939-f004:**
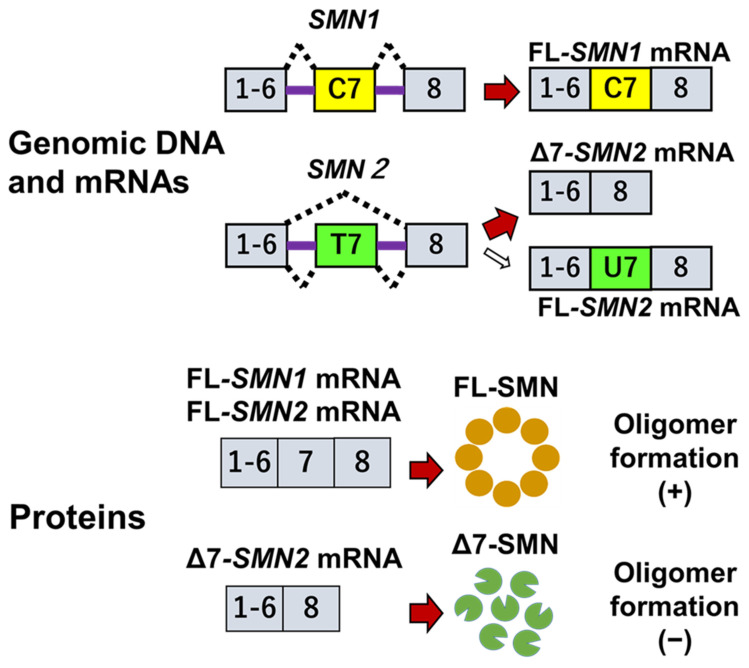
Alternative splicing of *SMN* genes and its protein products. When exon 7 is included in *SMN2* mRNA (full-length (FL)-*SMN2* mRNA), then a full-length SMN (FL-SMN) protein is produced. When exon 7 is excluded from *SMN2* mRNA (Δ7-*SMN2* mRNA), then a truncated SMN (Δ7-SMN) protein is produced. Δ7-SMN protein is unstable and almost nonfunctional because of its inability to form oligomers (lack of self-oligomerization).

**Figure 5 ijms-24-11939-f005:**
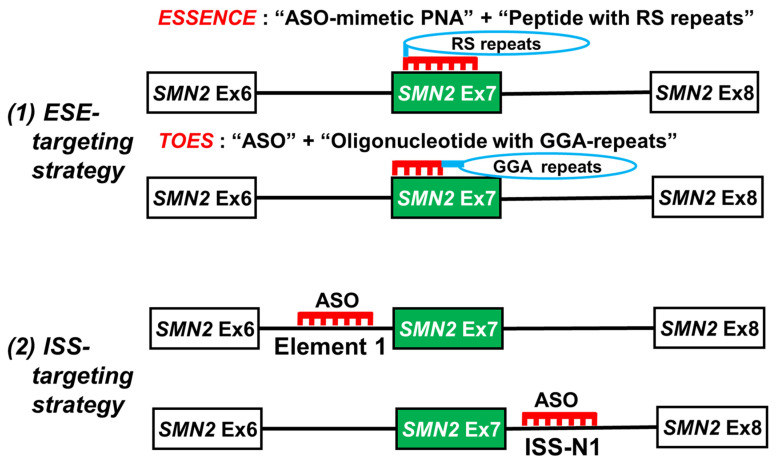
ESE-targeting and ISS-targeting strategies. To prevent skipping of *SMN2* exon 7, exonic splicing enhancer (ESE)-targeting and intronic splicing silencer (ISS)-targeting strategies were established. In 2003, two ESE-targeting compounds, namely exon-specific splicing enhancement by small chimeric effectors (ESSENCE) [102] and targeted oligonucleotide enhancers of splicing (TOES) [103], were devised. In 2002, a cis-acting element (element 1) that regulated *SMN2* exon 7 splicing was identified in intron 6 [104]. In 2006, another cis-acting element (ISS-N1) that regulated *SMN2* exon 7 splicing was identified in intron 7 [105]. Further details are provided in the text. ASO: antisense oligonucleotide, PNA: peptide nucleic acid.

**Figure 6 ijms-24-11939-f006:**
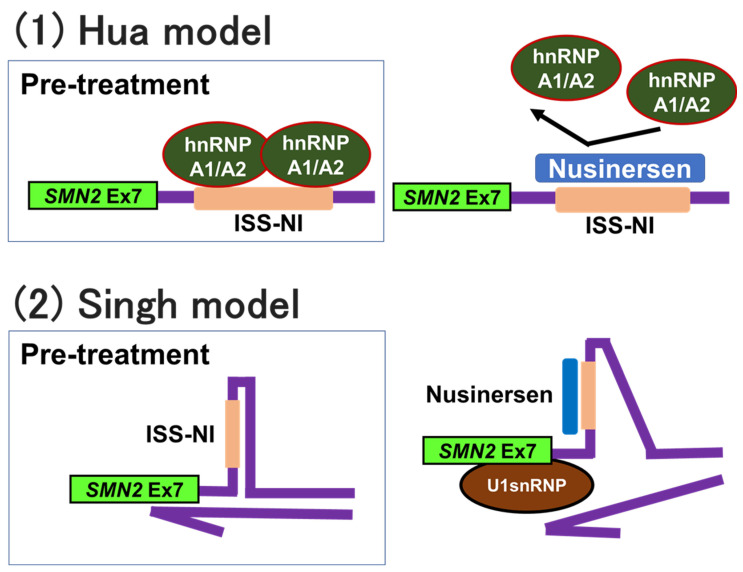
Mechanism of action of nusinersen. Hua et al. proposed that nusinersen hampers interaction between ISS-N1 in *SMN2* pre-mRNA intron 7 and hnRNP A1/A2 [106]. However, Singh et al. proposed a different mechanism. Nusinersen alters the secondary structure of *SMN2* pre-mRNA, which promotes recruitment of U1 snRNP at the 5′-splice site of *SMN2* pre-mRNA exon 7 [134].

**Figure 7 ijms-24-11939-f007:**
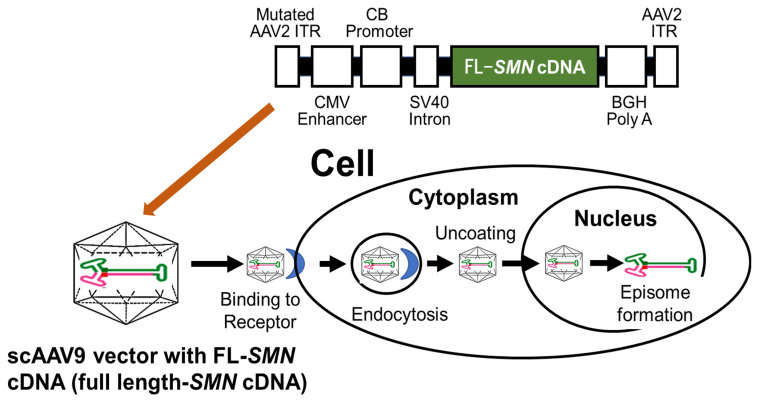
Mechanism of action of onasemnogene abeparvovec. Onasemnogene abeparvovec is an scAAV9 vector-based drug [140]. After entering the host cell, the scAAV vector translocates into the nucleus, where the transgene acts as an episome (a small, stable chromosome separate from the native chromosome). The scAAV ITR increases the speed at which the double-stranded transgene is transcribed and the resulting protein is produced. The hybrid CMV enhancer and CB promoter activates the transgene to allow for continuous and sustained SMN protein expression. Abbreviations are as follows, AAV2 (adeno-associated virus serotype 2); AAV9 (AAV serotype 9); BGH Poly A (bovine growth hormone polyadenylation); CB (chicken β-actin); cDNA (complementary DNA); CMV (cytomegalovirus); ITR (inverted terminal repeat); scAAV (self-complementary AAV); SV (simian virus).

**Figure 8 ijms-24-11939-f008:**
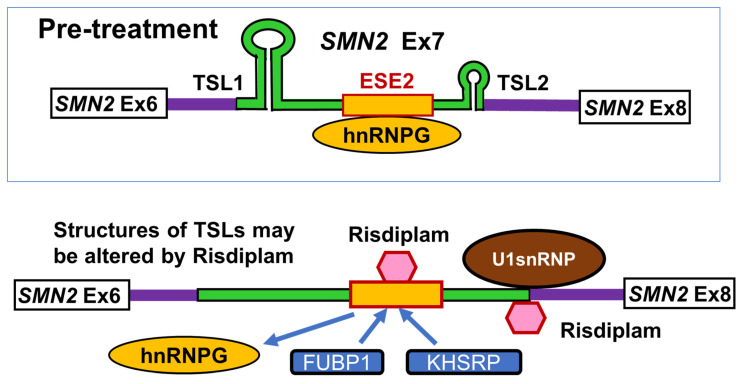
Mechanism of action of risdiplam. According to studies in which risdiplam analogues were used, risdiplam may act on two regions of *SMN2* exon 7 for highly selective pre-mRNA splicing [147,148]. The first region is the 5′-splice site of exon 7 (TSL2) of the pre-mRNA transcribed from *SMN2*. A risdiplam molecule stabilizes the duplex of the 5′-splice site RNA sequence and the U1 snRNP RNA sequence and promotes splicing initiation. The second region is the internal structure around exonic splicing enhancer 2 (ESE2) within *SMN2* exon 7. A risdiplam molecule binds to the ESE2 region, which alters the stem-loop structure (TSL1) in the first half of *SMN2* exon 7. The risdiplam molecule bound to the ESE2 region inhibits hnRNPG protein binding while promoting the binding of two splicing regulatory proteins: far upstream element binding protein 1 (FUBP1) and its homolog, KH-type splicing regulatory protein (KHSRP). TSL1 includes a cis-acting element that suppresses *SMN2* exon 7 splicing, and FUBP1 and KHSRP proteins are trans-acting factors that promote *SMN2* exon 7 splicing.

**Table 1 ijms-24-11939-t001:** Classification of SMA.

Type	Onset	Function	Median Survival	*SMN2* Copy Number in *SMN1*-Deleted Patients
0	Prenatal	Respiratory failure at birth	Weeks	1
I	0–6 months	Never sit	<1 years	2–3
II	<18 months	Sit	>25 years	3
III	>18 months	Stand or ambulatory	adult	3–4
IV	>30 years	Ambulatory	adult	≥4

Adapted from the original by Arnold et al. [11].

**Table 2 ijms-24-11939-t002:** FDA-approved drugs for SMA.

	Nusinersen	Onasemnogene Abeparvovec	Risdiplam
Drug type	Antisense oligonucleotide	Adeno-associated viral vector	Small molecular compound
Mechanism of action	Modification of *SMN2* pre-mRNA Splicing	*SMN* gene transfer	Modification of *SMN2* pre-mRNA Splicing
Administration	Intrathecal	Intravenous	Oral
FDA approved year	2016	2019	2020
Patient age	All	<2 years	>2 months *
Clinical trials involving symptomatic patients	ENDEAR CHERISH	START	FIREFISH SUNFISH
Clinical trials involving pre-symptomatic patients	NURTURE	SPR1NT	RAINBOWFISH

* Risdiplam was approved in the US for pediatric and adult patients with SMA of all ages in 2022.

## Data Availability

Not applicable.

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
