# Peer review of "Spinal Muscular Atrophy: The Past, Present, and Future of Diagnosis and Treatment"

_ijms, 2023, doi:10.3390/ijms241511939_

Round 1
Reviewer 1 Report
In the manuscript entitled “Spinal muscular atrophy: The past, present, and future of diagnosis and treatment”, the authors comprehensively reviewed the history and future directions of spinal muscular atrophy. Essentially, it is a well-written and comprehensive manuscript. Three therapeutic drugs and gene therapy have been approved, and newborn screening would become fruitful for many countries in the near future.
The reviewer would like to give several suggestions for improvement.
Comments
Figure 2 is too schematic. It is like a cartoon for children. A more scientific-oriented figure is proper.
Table 1 is adapted from the original publication in ref [11]. Do authors have permission?
Page 7, reconsider the title of 3.2.2. The reviewer thinks that authors like to show the expression changes in aging.
Figure 4, in proteins, do authors mean SMN2 in this figure?
Page 12, lines 487-490, please cite relevant and important papers for the 22 years from 1995 to 2016.
There are two Figure 7. Figure 7 in page 22 is likely to be Figure 8. In this figure, FUB1 is shown; however, FUBP1 is shown on the legend and the text. In Figure 6, It is for SMN2; however, no description is found in the figure and the legend.
Other comments
SMN is used to show both SMN1 and SMN2 rather randomly throughout the manuscript. Please re-write to clarify whether the authors show SMN1 or SMN2 more specifically.
Please check affiliation 6. There is no “6” in the authors.
Page 3, line 128, Type I should read type. I
Page 4, line 142-, please unify the parentheses and space (“ ”).
Page 7, line 280, remove the extra “-" after tyrosine.
Page 10, line 400, remove the extra “-“ after SMA.
Page 22, line 907, Four is duplicated. Line 923, endpoint is proper.
Page 23, line 973, With is duplicated.
References
Please check the format of refs 21-23.
Please see the comments for authors.
Author Response
Reviewer 1
(General comment)
In the manuscript entitled “Spinal muscular atrophy: The past, present, and future of diagnosis and treatment”, the authors comprehensively reviewed the history and future directions of spinal muscular atrophy. Essentially, it is a well-written and comprehensive manuscript. Three therapeutic drugs and gene therapy have been approved, and newborn screening would become fruitful for many countries in the near future. The reviewer would like to give several suggestions for improvement.
Answer: Thank you very much for your encouraging comments and useful suggestions. We followed your suggestions to improve our manuscript.
(Specific comment 1)
Figure 2 is too schematic. It is like a cartoon for children. A more scientific-oriented figure is proper.
Answer: Thank you very much for your suggestion. Following your suggestion, we replaced the figure with a more scientific-oriented figure.
(Specific comment 2)
Table 1 is adapted from the original publication in ref [11]. Do authors have permission?
Answer: Yes, we have got the permission to use the Table 1 in ref [11].
(Specific comment 3)
Page 7, reconsider the title of 3.2.2. The reviewer thinks that authors like to show the expression changes in aging.
Answer: Thank you very much for your suggestion. We changed the title from “Age-dependent manner in expression” to “High SMN expression in the fetal period”.
(Specific comment 4)
Figure 4, in proteins, do authors mean SMN2 in this figure?
Answer: Thank you very much for your suggestion. We added some words indicating FL-SMN1 mRNA, FL-SMN2 mRNA and △7-SMN2 mRNA.
(Specific comment 5)
Page 12, lines 487-490, please cite relevant and important papers for the 22 years from 1995 to 2016.
Answer: Thank you very much for your suggestion. As relevant and important papers for the 22 years from 1995 to 2016 are cited in the following subsections, we just put the following sentences at the end of this paragraph, enabling us to avoid the repetition of citation numbers.
“Medical advances in this period will be discussed in the following subsections, citing important relevant literatures.”
(Specific comment 6)
There are two Figure 7. Figure 7 in page 22 is likely to be Figure 8. In this figure, FUB1 is shown; however, FUBP1 is shown on the legend and the text.
Answer: Thank you very much for pointing out our mistakes. The Figure 7 in page 22 should be Figure 8. “FUB1” in the figure should be “FUBP1”.
(Specific comment 7)
In Figure 6, it is for SMN2; however, no description is found in the figure and the legend.
Answer: Thank you very much for pointing out our mistakes. We added “SMN2 Ex7” in the figure.
(Specific comment 8)
SMN is used to show both SMN1 and SMN2 rather randomly throughout the manuscript. Please re-write to clarify whether the authors show SMN1 or SMN2 more specifically.
Answer: Thank you very much for suggestion. We carefully checked the terms. As for proteins when we discuss the SMN protein in this manuscript, we differentiate between full-length and truncated (△7) proteins. But we do not differentiate between the SMN1 gene product and the SMN2 gene product. However, following the suggestion of Reviewer 1, to avoid confusion of the gene name and the protein name, we used “SMN protein” in the revised version. As for genes, we used SMN1 and SMN2. However, as for AAV vector drug, the sequence of the constructed SMN cDNA has never been published. Thus, we just describe SMN cDNA.
(Specific comment 9)
Please check affiliation 6. There is no “6” in the authors.
Page 3, line 128, Type I should read type. I
Page 4, line 142-, please unify the parentheses and space (“ ”).
Page 7, line 280, remove the extra “-" after tyrosine.
Page 10, line 400, remove the extra “-“after SMA.
Page 22, line 907, Four is duplicated. Line 923, endpoint is proper.
Page 23, line 973, With is duplicated.
Answer: Thank you very much for pointing out our errors. Following your suggestion, we corrected all the errors you pointed out. We really appreciate your careful check of our manuscript.
Reviewer 2 Report
The review is very detailed, covers all aspects of SMA from the discovery and the first descriptions of this disease to the present. The purpose of writing this work is not very clear to me and its authors are not explicitly formulated.
A lot of review papers and recommendations containing the same information are available for this disease.
Major remarks:
1) The authors have not presented an algorithm for the selection of works and articles included in the work
2) Figure 2 contains an inaccuracy. The SMN1 and SMN2 genes have the same purpose.
3) It is definitely necessary to keep the part about treatment. Systematization of information is necessary.
4) It is necessary to shorten the discussion section. It completely repeats the main text.
Minor remarks:
1) Line 421 states "In our study, the mean SMN2 copy number in patients with SMA (including all sub- 421 types) was higher than that in control subjects " - is this really a typo anyway?
2) There is no risdisplam in the drug development section. In the next section, it appears among the drugs approved by the FDA. I propose to merge the sections concerning therapy into one for a better understanding.
3) The work is entitled "Spinal muscular atrophy: The past, present, and future of diag- 2 nosis and treatment". It remains unclear to me what the authors see in the future of treatment and diagnosis of the disease?
4)It is necessary to discuss small mutations of the SMN1 gene at least a little.
I believe that this review needs substantial revision and reduction.
Author Response
Reviewer 2
(General comment 1)
The review is very detailed, covers all aspects of SMA from the discovery and the first descriptions of this disease to the present. The purpose of writing this work is not very clear to me and its authors are not explicitly formulated.
Answer: As you can see, this is a narrative review, not a systematic review.
A systematic review is a protocol driven comprehensive review and synthesis of data focusing on a topic or on related key questions. Clearly defined research question may be essential in writing a systematic review. From this point of view, you may think that that there are no clearly defined research questions in this manuscript, and we are not explicitly formulated.
We believe that narrative reviews are best suited for historical analysis of advance in SMA research. Preliminary studies with no solid data would be discarded in a systematic review process. However, in a narrative review, we can pick up such preliminary studies with a great idea.
(General comment 2)
A lot of review papers and recommendations containing the same information are available for this disease.
Answer: We agree with you that the review articles published recently share the same information. However, we found that many of them lack the historical viewpoint of the disease. We want to document the history from the first case of SMA to the most recent advance in this field.
(Specific comment 1)
The authors have not presented an algorithm for the selection of works and articles included in the work.
Answer: This is a narrative review article, not a systematic review. In a narrative review, we do not think it is necessary to show the literature selection algorithm.
(Specific comment 2)
Figure 2 contains an inaccuracy. The SMN1 and SMN2 genes have the same purpose.
Answer: Thank you very much for your suggestion. We show two models of the alignment (or direction) of SMN1 and SMN2.
(Specific comment 3)
It is definitely necessary to keep the part about treatment. Systematization of information is necessary.
Answer: Thank you very much for your suggestion. We have systematically shown the current treatments in subsections 6 to 8. We have explained FDA-approved drugs in subsection 6, newborn screening in subsection 7, and problems in the current treatment in subsection 8.
(Specific comment 4)
It is necessary to shorten the discussion section. It completely repeats the main text.
Answer: Thank you very much for your suggestion. Following your suggestion, we completely deleted the repeat part of the main text, that is, “a sketch of SMA history” part.
(Specific comment 5)
Line 421 states "In our study, the mean SMN2 copy number in patients with SMA (including all subtypes) was higher than that in control subjects " - is this really a typo anyway?
Answer: The description is right. It is not typo. Gene conversion events which often occurred in SMA patients increased the mean SMN2 copy number of the patients.
(Specific comment 6)
There is no risdisplam in the drug development section. In the next section, it appears among the drugs approved by the FDA. I propose to merge the sections concerning therapy into one for a better understanding.
Answer: Thank you very much for your suggestion. SMN-C3, that is written in the drug development section, was the prototype of risdiplam. But we did not clearly say it in the original version. Following your suggestion, we added a sentene in the revised version, “SMN-C3 may be considered the prototype of risdiplam.”in the revised version.
(Specific comment 7)
The work is entitled "Spinal muscular atrophy: The past, present, and future of diagnosis and treatment". It remains unclear to me what the authors see in the future of treatment and diagnosis of the disease?
Answer: Newborn screening followed by treatment with new drugs will become a standard of diagnosis and treatment for SMA in the very near future.
In the Abstract and Conclusion sections, we wrote clearly, “Early detection by newborn screening and early treatment with new drugs are expected to soon become the standards in the field of SMA.”
Besides, in the Discussion section, we also explained the possible treatment other than the current treatment with three drugs (nusinersen, onasemnogene abeparvovec, risdiplam)
(Specific comment 8)
It is necessary to discuss small mutations of the SMN1 gene at least a little.
Answer: Thank you very much for your suggestion. In this manuscript, small mutations of the SMN1 gene are referred to as intragenic SMN1 mutations. Following your suggestion, we added (a small mutation in SMN1 gene) to the first appearance of “an intragenic SMN1 mutation.”.
(Specific comment 9)
I believe that this review needs substantial revision and reduction.
Answer: Thank you for your feedback. Following your suggestion, we deleted some parts in the revised version.
Reviewer 3 Report
The manuscript of Nishio et al. is a comprehensive review of Spinal muscular atrophy, showing not only the recent changes in the therapy but also the historical background and discoveries that led to the diagnostics and therapy. The authors describe a range of topics, including phenotypic variation, classification, genetic modifiers, the development of therapies, clinical trials in pre-symptomatic and symptomatic patients, problems associated with new therapies, such as switching therapies, and recent issues, such as newborn screening. The authors divide their manuscript into three periods: description of phenotypes to chromosomal mapping, the second period from identification of the SMN protein to development of the drugs, and the first period starting from FDA-drug approval. This division facilitates understanding a long history of work on the SMA.
the manuscript is very important and should be published. However, I have some comments, given the complexity of the manuscript.
Comments:
Tables, especially regarding the available drugs and clinical studies, would facilitate the understanding. Also, a graphic with a map of neonatal screening availability would be helpful.
Line 43 the phenotype of the first patients is mentioned, it would be good to describe it in a few words here
Section 3.1 Only SMN1 is mentioned, but in the figure, there is also SMN2. Either the figure should be changed or SMN2 described shortly in the text, not only in the Figure.
Lines 294-295 On the other hand, endocytosis impairment in synaptic transmission at NMJs has recently been attracting attention - I would give examples, not only references here
In the paragraph on SMN2, I would add information about number of copies across different populations
Subtitle 5.1. Exciting Days with one goal in mind-I would change it to the more informative title
Paragraph 5.4.2. VPA-I have the feeling that lines 592-602 do not fit here
Paragraph 7.4 - please list alo other challenges to the neonatal SMA screening, not only high drug cost
Paragraph 8.4 Please describe also situation where more than one drug is given in parallel and the background of it
Author Response
Reviewer 3
(General comment)
The manuscript of Nishio et al. is a comprehensive review of Spinal muscular atrophy, showing not only the recent changes in the therapy but also the historical background and discoveries that led to the diagnostics and therapy. The authors describe a range of topics, including phenotypic variation, classification, genetic modifiers, the development of therapies, clinical trials in pre-symptomatic and symptomatic patients, problems associated with new therapies, such as switching therapies, and recent issues, such as newborn screening. The authors divide their manuscript into three periods: description of phenotypes to chromosomal mapping, the second period from identification of the SMN protein to development of the drugs, and the first period starting from FDA-drug approval. This division facilitates understanding a long history of work on the SMA.
The manuscript is very important and should be published. However, I have some comments, given the complexity of the manuscript.
Answer: Thank you very much for your encouraging comments and useful suggestions. We followed your suggestions to improve our manuscript.
(Specific Comment 1)
Tables, especially regarding the available drugs and clinical studies, would facilitate the understanding. Also, a graphic with a map of neonatal screening availability would be helpful.
Answer: Thank you very much for your suggestion. We added a table of FDA-approved Drugs for SMA in the revised manuscript. Regarding a map of neonatal screening availability in the world, we cited an article by Dangouloff et al. (2021). According to their data, neonatal screening for SMA was available in Taiwan, USA, Germany, Belgium, Australia, Italy, Russia, Canada, and Japan, as of December 29, 2020.
(Specific Comment 2)
Line 43 the phenotype of the first patients is mentioned, it would be good to describe it in a few words here.
Answer: Thank you very much for your suggestion. We added the following sentence, “It is worthy of notice that their phenotypes were consistent with those of SMA type II, according to the current classification of SMA (we will discuss it later).”
(Specific Comment 3)
Section 3.1 Only SMN1 is mentioned, but in the figure, there is also SMN2. Either the figure should be changed or SMN2 described shortly in the text, not only in the Figure.
Answer: Thank you very much for your suggestion. We added short description about SMN2. In addition, we revised the figure.
(Specific Comment 4)
Lines 294-295 On the other hand, endocytosis impairment in synaptic transmission at NMJs has recently been attracting attention - I would give examples, not only references here.
Answer: Thank you very much for your suggestion. We explained the achievements of Hosseinibarkooie et al. and Riessland et al.
(Specific Comment 5)
In the paragraph on SMN2, I would add information about number of copies across different populations
Answer: Thank you very much for your suggestion. We added a new subsection, 4.3.3. Variation of SMN1 gene copy numbers in different populations.
(Specific Comment 6)
Subtitle 5.1. Exciting Days with one goal in mind-I would change it to the more informative title
Answer: Thank you very much for your suggestion. Following your suggestion, we changed the subtitle from “Exciting Days with one goal in mind” to “Development of therapeutic options for SMA”.
(Specific Comment 7)
Paragraph 5.4.2. VPA-I have the feeling that lines 592-602 do not fit here
Answer: Thank you very much for your suggestion. Following your suggestion, we deleted the sentences of lines 592-602 in the original version.
(Specific Comment 8)
Paragraph 7.4 - please list altogether challenges to the neonatal SMA screening, not only high drug cost
Answer: Thank you very much for your suggestion. Following your suggestion, we added two things low public awareness of SMA and stigmatization against a genetic disorders.
(Specific Comment 9)
Paragraph 8.4 Please describe also situation where more than one drug is given in parallel and the background of it
Answer: Thank you very much for your suggestions. Following your suggestion, we added some paragraphs in the revised version.
(1) About switching from splicing modifiers to onasemnogene abeparvovec (subsection 8.5 in the revised version)
(2) About combination therapy of onasemnogene abeparvovec and splicing modifiers (subsection 8.6 in the revised version)
(3) About combination therapy of splicing modifiers and existing drug like VPA (subsection 8.6 in the revised version)
Reviewer 4 Report
The presented analytical review covers the very important and interesting area of the clinical medicine and human genetics and pharmacology. The SMA has been a hightmare of the world clinical medicine for many years. The authors give the detailed history of the clinical studies, genetic analysis and search for the apropriate and efficient drugs for this fatal diseases. The revies is well-composedd, well-written and complete in all areas regarding clinical science, genetics and pharmacology of SMA. Anyway the r3eview article could be better if it describe the experimental works on state of the skeletal muscle in SMA. If the authors consider that the muscle story is outside from their intensions they should tell this statement in the Introduction.
Author Response
Reviewer 4
(General comment)
The presented analytical review covers the very important and interesting area of the clinical medicine and human genetics and pharmacology. The SMA has been a nightmare of the world clinical medicine for many years. The authors give the detailed history of the clinical studies, genetic analysis and search for the appropriate and efficient drugs for this fatal disease. The review is well-composed, well-written and complete in all areas regarding clinical science, genetics and pharmacology of SMA. Anyway, the review article could be better if it describes the experimental works on state of the skeletal muscle in SMA. If the authors consider that the muscle story is outside from their intensions they should tell this statement in the Introduction.
Answer: Thank you very much for your encouraging comments. I agree with you on the recent progress of the muscle study in SMA field. Following your suggestion, we added the experimental works on state of the skeletal muscle in the revised version.
Round 2
Reviewer 1 Report
The manuscript seems properly revised.
After acceptance, the reviewer recommends to read through the manuscript and correct properly.
For example, the Italic "is" in the third line in the Figure 2 legend should not be Italic.
Please check the format of refs 21-23.
Reviewer 3 Report
Thank you for the revisions. It is an important contribution to the field.